# ARM: Discovering Agentic Reasoning Modules for Generalizable Multi-Agent Systems

## Abstract

Large Language Model (LLM)-powered Multi-agent systems (MAS) have achieved state-of-the-art results on various complex reasoning tasks. Recent works have proposed techniques to automate the design of MASes, eliminating the need for manual engineering. However, these techniques perform poorly, often achieving similar or inferior performance to simple baselines. Furthermore, they require computationally expensive re-discovery of architectures for each new task domain and expensive data annotation on domains without existing labeled validation sets. A critical insight is that simple Chain of Thought (CoT) reasoning often performs competitively with these complex systems, suggesting that the fundamental reasoning unit of MASes, CoT, warrants further investigation. To this end, we present a new paradigm for automatic MAS design that pivots the focus to optimizing CoT reasoning. We introduce the **A**gentic **R**easoning **M**odule (ARM), an agentic generalization of CoT where each granular reasoning step is executed by a specialized reasoning module. This module is discovered through a tree search over the code space, starting from a simple CoT module and evolved using mutations informed by reflection on execution traces. The resulting ARM acts as a versatile reasoning building block which can be utilized as a direct recursive loop or as a subroutine in a learned meta-orchestrator. Our approach significantly outperforms both manually designed MASes and state-of-the-art automatic MAS design methods. Crucially, MASes built with ARM exhibit superb generalization, maintaining high performance across different foundation models and task domains without further optimization.

## 1 Introduction

Chain-of-thought (CoT) prompting has emerged as one of the most effective techniques for eliciting complex reasoning from Large Language Models (LLMs) (Wei et al., 2022). By instructing models to generate a series of intermediate steps that lead to a final answer, CoT significantly enhances performance on tasks requiring arithmetic, commonsense, and symbolic reasoning (Nye et al., 2021; Kojima et al., 2022). This simple yet powerful method allows LLMs to break down complex problems into more manageable sub-problems, effectively externalizing the reasoning process over a sequence of generated tokens before arriving at a solution (Wei et al., 2022; Yao et al., 2023a). Recent advancements have also extended CoT with formal verification and multi-agent perspectives, such as MA-LoT (Wang et al., 2025).

Building on the capabilities of individual LLMs, Multi-Agent Systems (MAS) have recently achieved state-of-the-art results on complex reasoning benchmarks (Park et al., 2023; Qian et al., 2023; Hong et al., 2023). These systems typically consist of multiple LLM-powered agents, each assigned a specific role or expertise, orchestrated by a meta-agent or a predefined communication protocol (Wu et al., 2023; Li et al., 2023). While the collaborative nature of MAS enables division of labor and synthesis of diverse perspectives (Chen et al., 2023; Dong et al., 2023), recent work has shifted toward the automatic construction of such systems. Emerging automatic MAS generation frameworks demonstrate how agent roles, communication protocols, and workflows can be synthesized directly by LLMs without manual design. For instance, FlowReasoner and AFlow illustrate this trend by automatically generating agent roles and workflows for LLM-based systems, reducing the need for manual design (Zhang et al., 2025c; Kim et al., 2024).

Although MAS approaches have consistently pushed the boundaries of performance, recent studies have revealed a surprising trend: in many cases, a well-prompted single-agent CoT baseline can outperform or perform on par with these complex, multi-agent architectures (Wang et al., 2024; Yao & Yadav, 2025). We also show these observations in our results (Table (1) This finding is significant, as CoT is one of the foundational techniques for LLM reasoning. Its continued competitiveness suggests that the core reasoning unit—the individual thought or step—is of paramount importance. Arguably, the majority of recent research efforts have been dedicated to designing more elaborate MAS frameworks, while the fundamental CoT baseline has remained largely unchanged (Creswell et al., 2022; Chen et al., 2024). Our work pivots from this trend to focus on fundamentally reshaping and enhancing the CoT paradigm for the agentic era by redefining the nature of each reasoning step.

In this work, we introduce the Agentic Reasoning Module (ARM), a novel sequential reasoning approach where each granular step is executed by a specialized, self-contained reasoning agent. The core motivation is to elevate the "thinking" steps of CoT from simple textual continuation to the execution of a sophisticated, agentic block. This block is not manually designed but is instead automatically discovered through an evolutionary process. Starting with a basic CoT procedure, the module is iteratively mutated and refined based on its performance on a generic validation dataset of reasoning problems, resulting in a robust and versatile reasoning procedure that can be applied recursively at each step of solving a challenging multi-step problem.

The prevailing paradigm for MAS design often leads to systems that are highly domain-specific, with individual agents meticulously tuned for particular skills or tasks (Hu et al., 2025; Zhang et al., 2025b). While single-agent systems are generally considered more versatile, they too are often optimized for a narrow set of domains (LaMDAgent, 2025; ScribeAgent, 2024). In contrast, our work focuses on enhancing the universally applicable CoT framework. The agentic block within ARM can be optimized on any generic domain, yielding a general-purpose reasoning technique analogous to the original CoT. We demonstrate that this approach not only achieves superior performance but also exhibits greater generalizability. As we show, MAS built with ARM significantly outperform prominent MAS approaches across diverse agentic datasets without domain-specific tuning.

Our methodology uses the simple yet powerful CoT as a starting seed for the evolutionary discovery of ARM. A meta-agent orchestrates this process, performing a tree search over the code space of possible reasoning modules. Mutations and evolutions are guided by a reflection mechanism that analyzes execution traces from previous attempts, identifying weaknesses and proposing targeted improvements. Furthermore, this meta-agent discovers global strategies to orchestrate collaborations between parallel ARM reasoning traces, effectively creating a high-performance MAS from optimized, homogeneous building blocks. Overall, our work underscores the immense potential of evolving fundamental reasoning methodologies like CoT, presenting a more robust and scalable alternative to the development of increasingly complex and fragile heterogeneous MAS systems. Key contributions of our work are as follows:

- We present the Agentic Reasoning Module (ARM), an evolved and enhanced version of Chain-of-Thought reasoning. We demonstrate that systems built with ARM substantially outperform existing manually designed and automatically discovered multi-agent systems on complex reasoning tasks.

- We show that ARM is a significantly more generalizable reasoning module. MAS constructed with ARM maintain high performance across different underlying foundation models and task domains without requiring re-optimization, highlighting its robustness.

- We provide a rigorous justification and detailed ablations on the validity of our training objective demonstrating the effectiveness of the proposed MAS discovery strategy.

## 2 RELATED WORKS

**Single-Agent and Multi-Agent Reasoning Systems**   The landscape of LLM-based reasoning is broadly divided into single-agent and multi-agent paradigms. Single-agent systems have demonstrated remarkable capabilities by augmenting the core LLM with sophisticated reasoning and action frameworks. A prominent example is the ReAct framework, which interleaves reasoning steps with actions, enabling the agent to interact with external tools like search engines to gather information and refine its reasoning process (Yao et al., 2023b). Other approaches have focused on enhancing

single agents with self-reflection and memory to learn from past mistakes and improve performance iteratively (Shinn et al., 2023; Madaan et al., 2023). While these systems are powerful, their development has often focused on narrower tasks, such as tool-based search, retrieval, and question answering, rather than general-purpose complex reasoning.

In parallel, Multi-Agent Systems (MAS) have emerged as a dominant approach for tackling highly complex problems, often outperforming single-agent counterparts (Park et al., 2023; Qian et al., 2023). Frameworks like AutoGen (Wu et al., 2023), Camel (Li et al., 2023), and MetaGPT (Hong et al., 2023) orchestrate multiple LLM-powered agents, each assigned a specialized role (e.g., programmer, critic, tester). These agents collaborate, debate, and synthesize information to produce solutions for tasks like software development and complex reasoning. A key characteristic of these systems is their heterogeneous nature; each agent is distinct, with a manually engineered role and persona, connected through a predefined and often complex communication topology. In stark contrast, our ARM-based approach constructs a powerful MAS from homogeneous building blocks. The ARM itself is a self-contained, versatile reasoning module that is applied repeatedly, acting as the fundamental unit of thought for all "agents" in the system, thereby simplifying the design while enhancing generalizability.

**The Surprising Efficacy of Simple Reasoning Baselines**   Despite the architectural complexity of many state-of-the-art MAS, a critical and recurring observation is the surprising competitiveness of simple reasoning baselines (Dubey, 2023). Foundational techniques like Chain-of-Thought (CoT) (Wei et al., 2022), and simple extensions like Self-Consistency (CoT-SC) which samples multiple reasoning chains and takes a majority vote (Wang et al., 2022), often achieve performance on par with, or even superior to, intricate multi-agent frameworks (Zhang et al., 2025a). This phenomenon is particularly pronounced with the advent of increasingly powerful frontier foundation models (Ke et al., 2025). As these models develop stronger native reasoning abilities, the high-level conceptual guidance provided by a simple CoT prompt is often sufficient to unlock their full potential, rendering the overhead of complex agent orchestration less impactful. This suggests that the primary bottleneck is not necessarily the high-level orchestration strategy but the quality and robustness of the fundamental, step-by-step reasoning process. Our work is directly motivated by this insight, positing that evolving the core reasoning operator—the "thought" in the chain—is a more fruitful direction than designing ever-more-complex superstructures around a static, simple CoT unit.

**Automated Design of Multi-Agent Systems**   Recognizing the significant manual effort required to design effective MAS, recent research has explored automating this process. Approaches like ADAS (Hu et al., 2025), Aflow Zhang et al. (2025b), and Flow-Reasoner (Gao et al., 2025) aim to automatically discover the optimal agent roles and their interaction topology for a given task domain. However, these techniques suffer from two major drawbacks. First, they are computationally expensive, requiring a costly re-discovery process for each new task domain. Second, the discovered systems are often highly specialized and brittle, tuned specifically for the validation data of a single domain. As our results will later show, with the latest generation of foundation models, these automatically discovered systems can be outperformed by simple CoT baselines. Our work diverges from this paradigm. Instead of discovering a complex, domain-specific agent topology, we focus on discovering a single, domain-agnostic reasoning module (ARM). This ARM acts as a universal, high-quality building block that provides superior performance and generalizability without the need for task-specific rediscovery, offering a more scalable and robust path forward for MAS design.

**LLM based Prompt Optimizers**   Recent research has focused on LLMs as prompt optimizers, leveraging their generative and reasoning capabilities to automatically improve prompts within a fixed workflow Zhou et al. (2023); Yang et al. (2024); Khattab et al. (2024); Guo et al. (2024); Novikov et al. (2025); Fernando et al. (2024). Notably, evolutionary approaches coupled with deep reflection over rollouts, such as in GEPA Agrawal et al. (2025), have been shown to offer significant advantages in sample efficiency compared to methods that involve updating model weights via Reinforcement Learning.

## 3   METHODOLOGY: DISCOVERING THE AGENTIC REASONING MODULE

We introduce the **Agentic Reasoning Module (ARM)**, a self-contained, code-based multi agentic system designed to execute a single, granular step within a complex reasoning process. ARM is conceived as a structured, agentic replacement for a single step in a Chain of Thought (CoT) sequence Wei et al. (2022). While standard CoT prompts an LLM to generate the next reasoning step

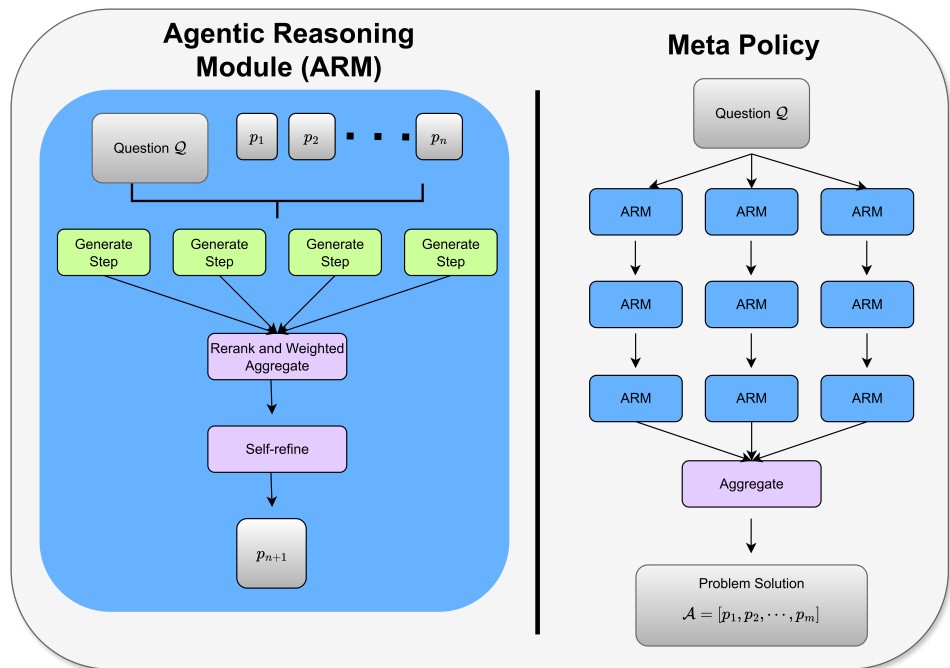

Figure 1: An illustration of the proposed ARM module on the left and the meta policy on the right using "Self refine" as an example MAS. The ARM module takes a question and previous reasoning steps and executes a MAS to get the next step. The meta policy uses ARM as a sub-module and orchestrates the overarching global strategy. Note that this is for illustration only, the actual step generator and the meta policy discovered by Algorithm-1 is more complex (See Appendix).

via naive, monolithic textual generation, an ARM employs an internal multi-agent system (MAS) to produce reasoning steps with greater structure and control.

Following prior work, Hao et al. (2023); Zhang et al. (2024), we define the multi-agentic system as a programming module - a self contained Python function block, while allowing for structured control flow and access to essential APIs such as calling an external LLM, structuring the role and the prompt, and input/output format expectations. Functionally, an ARM accepts the initial problem statement and prior reasoning steps as input, and continues the reasoning until the next logical step in the solution.

### 3.1 A Decomposable Framework for Agentic Reasoning

Let the distribution over problem-solution pairs be $\mathcal{D}$ over $(\mathcal{Q}, \mathcal{A})$. A solution $\mathcal{A}$ consists of a sequence of reasoning steps $[p_1, p_2, \ldots, p_N]$, where each step $p_i$ belongs to the space of all possible reasoning steps $\mathcal{P}$. We model the problem-solving process with two key functions:

* **The Step-Generator Module ($m \in \mathcal{M}$):** This is a program that performs a single step of reasoning. It takes the problem question $q \in \mathcal{Q}$ and the history of previous reasoning steps $p_{in} \in \mathcal{P}^*$ as input and returns the next reasoning step $p_{out} \in \mathcal{P}$. Its signature is $m : \mathcal{Q} \times \mathcal{P}^* \to \mathcal{P}$. An **Agentic Reasoning Module (ARM)** is a structured, code-based implementation of such a module, which can itself be a self-contained MAS.

* **The Meta-Policy ($\pi \in \Pi$):** This is a higher-order program that defines the overarching strategy. It takes a question $q$ and a specific step-generator module $m$ and orchestrates calls to $m$ to generate a complete solution $a \in \mathcal{A}$. Its signature is $\pi : \mathcal{Q} \times \mathcal{M} \to \mathcal{A}$.

Within this framework, standard Chain of Thought (CoT) can be seen as a simple baseline pairing. It uses a basic step-generator, $m_{CoT}$, which is a single call to an LLM, and a simple **recursive meta-policy**, $\pi_{Rec}$, which applies $m_{CoT}$ repeatedly until a final answer is produced. Our approach independently discovers a more powerful module $m^*$ (the ARM) and a more sophisticated meta-policy $\pi^*$.

## 3.2 Discovering the Optimal Step-Generator ($m^*$)

Our primary goal is to find a step-generator module $m^*$ that is a general-purpose and superior replacement for the simple text generation step in $m_{CoT}$.

We can formalize a single reasoning step as an update function, $U_{m,q}$, that appends the output of module $m$ to the current reasoning history $h$:

$$U_{m,q}(h) = h \cdot [m(q, h)]$$

where $\cdot$ denotes list concatenation. A full, $n$-step reasoning trace generated by the recursive policy $\pi_{Rec}$ is thus the $n$-fold composition of this update function: $\pi_{Rec}(q, m) = U^n_{m,q}(\emptyset)$.

Ideally, we would discover the optimal module $m^*$ by maximizing the expected reward $\mathcal{R}$ over the entire problem-solving trace:

$$m^* = \underset{m \in \mathcal{M}}{\operatorname{argmax}} \quad \mathbb{E}_{(q,a) \sim \mathcal{D}} \left[ \mathcal{R}\left( \pi_{Rec}\left( q, m \right), a \right) \right]$$

However, optimizing this objective directly is intractable due to two main challenges: 1. **Difficult Credit Assignment:** The reward is observed only at the end of a long sequence of steps, making it difficult to determine which specific application of $m$ was responsible for the final outcome. 2. **Unconstrained Search Space:** The space of possible code-based modules $\mathcal{M}$ is vast, making an unguided search highly inefficient.

To address this, we introduce a practical **scaffolded surrogate objective**. Instead of evaluating $m$ on a full rollout generated by itself, we evaluate it within the stable context of a reference trace generated by the baseline $m_{CoT}$. Specifically, we replace a small, contiguous block of $l$ steps within an $n$-step CoT trace with our candidate module $m$. The optimization problem becomes:

$$m^* = \underset{m \in \mathcal{M}}{\operatorname{argmax}} \quad \mathbb{E}_{(q,a) \sim \mathcal{D}} \left[ \mathcal{R}\left( U^{n-l-i}_{m_{CoT},q} \circ U^l_{m,q} \circ U^i_{m_{CoT},q}(\emptyset), a \right) \right]$$

where $n = |\pi_{Rec}(q, m_{CoT})|$ is the length of the reference CoT trace, and the starting index $i$ is chosen randomly from $[0, n-1]$. This formulation isolates the performance contribution of $m$ to a small window, enabling direct credit assignment. Furthermore, the surrounding CoT context provides a powerful inductive bias, constraining the search to modules that behave as effective, incremental reasoning steps. In our experiments, we find $l = 3$ works well, as it is long enough to expose the module $m$ to critical compositional patterns—$(U_{m_{CoT},q} \circ U_{m,q})$, $(U_{m,q} \circ U_{m,q})$, and $(U_{m,q} \circ U_{m_{CoT},q})$—while keeping the optimization tractable.

## 3.3 Discovering the Optimal Meta-Policy ($\pi^*$)

While an optimized step-generator $m^*$ improves the quality of each reasoning step, the high-level strategy $\pi$ that orchestrates these steps is equally critical. A simple recursive policy, $\pi_{Rec}$, may be suboptimal for complex problems that could benefit from strategies like parallel rollouts (for self-consistency) or iterative refinement loops Wang et al. (2023); Madaan et al. (2023).

Searching for an optimal meta-policy $\pi^*$ by repeatedly evaluating candidates with the full, complex $m^*$ module is computationally prohibitive. Therefore, we adopt a surrogate-based approach here as well. We search for the optimal meta-policy $\pi^*$ using the fast and computationally cheap baseline step-generator, $m_{CoT}$, as a stand-in for $m^*$.

This zero-shot transfer from $m_{CoT}$ to $m^*$ is effective because our step-generator optimization process (Section 3.2) is explicitly designed to produce an $m^*$ that functions as a superior, "drop-in" replacement for $m_{CoT}$. A meta-policy that effectively orchestrates the simple steps of $m_{CoT}$ is thus highly likely to generalize to orchestrating the more powerful, but functionally analogous, steps of $m^*$. This allows us to efficiently explore the space of strategies, discovering sophisticated control flows like branching for parallel thought generation or conditional loops for verification, without incurring the high computational cost of using $m^*$.

## 3.4 Reflection-Guided Evolutionary Search

We discover both the optimal step-generator $m^*$ and meta-policy $\pi^*$ using a unified **Reflection-Guided Evolutionary Search** algorithm. This algorithm performs a tree search over the programmatic space of valid Python modules, where each node in the tree represents a specific program.

The search begins with a root node representing the baseline program ($m_{CoT}$ for the step-generator search and $\pi_{Rec}$ for the meta-policy search). The search then iteratively performs three steps:

1. **Selection:** A parent node (program) $p_{parent}$ is sampled from the current tree $\mathcal{T}$ using temperature sampling based on it's validation performance.

2. **Expansion:** A new child program is generated by a **Reviewer Agent**, an LLM-based agent that reflects on the parent program's execution traces, correctness, and mutation history to propose a targeted code modification.

3. **Evaluation:** The newly generated program is evaluated to obtain its average reward $\bar{\mathcal{R}}$. For a step-generator module, we use the scaffolded objective from Section 3.2. For a meta-policy, we evaluate its performance on a full problem rollout using $m_{CoT}$ as the step-generator.

This entire process is summarized in Algorithm 1.

### 3.4.1 THE REVIEWER AGENT

The expansion step is driven by a two-stage **Reviewer Agent** that intelligently mutates existing programs. This agent consists of two LLM-based components:

**Critic:** The Critic analyzes execution traces from the parent program. It identifies logical errors, inefficiencies, or patterns of failure, providing a concise, natural-language analysis of the program's strengths and weaknesses.

**Designer:** The Designer acts as the mutation operator. It takes the original program's code, its performance history, and the Critic's analysis as input. Based on this information, it proposes a single, targeted code modification aimed at addressing the identified issues, generating a complete, syntactically valid Python class for the new program.

This reflection-driven process ensures that the search evolves programs purposefully, rather than through random mutations, leading to more efficient discovery of high-performance modules and policies. The prompts used for the Critic and Designer are detailed in the Appendix.

## 4 EXPERIMENTS

### 4.1 BENCHMARKS

We evaluated our baselines and approach on multiple complex reasoning datasets. To assess complex mathematical reasoning capabilities, we utilized widely studied *American Invitational Mathematics Examination* (*AIME*[1]) and the *Harvard-MIT Mathematics Tournament* (*HMMT*[2]) datasets. For reasoning evaluations on specialized scientific knowledge, we used *GPQA*, a benchmark containing graduate-level questions in physics, chemistry, and biology designed to be challenging even for human experts (Rein et al., 2023). Finally, to measure practical, up-to-date reasoning and robustness against data contamination, we used *LiveBench Reasoning* [3], a dynamic benchmark with continuously evolving questions (Kaddour et al., 2023).

### 4.2 MODELS

We use OpenAI's o4-mini-high reasoning model as the MAS designer for both the baselines ADAS, AFlow, and our method ARM, as MAS generation requires frontier performance in coding, and instruction following. During validation and inference, we three models as backbone LLMs executing the MAS: two closed source models GPT-4.1-nano, GPT-4o and one open source model Llama-3.3-70B.

### 4.3 BASELINES

We compare our methodology against two distinct groups of multi-agent systems (MAS) baselines: popular handcrafted MAS systems and leading automated MAS generation approaches.

---

[1]`https://huggingface.co/datasets/MathArena/aime_2025`

[2]`https://huggingface.co/datasets/MathArena/hmmt_feb_2025`

[3]`https://huggingface.co/datasets/livebench/reasoning`

Handcrafted Multi-Agent Systems: We compare against several strong reasoning baselines. **Chain of Thought (CoT)** serves as the fundamental baseline, solving tasks through iterative textual reasoning. **CoT-Self Consistency (CoT-SC)** improves upon CoT by generating $n = 12$ parallel reasoning rollouts and selecting the final answer via a majority vote. **Self-Refine** employs a feedback loop where a Large Language Model (LLM) iteratively critiques and refines its own output. Lastly, **LLM-Debate** initializes multiple LLM agents with diverse roles to generate different reasoning paths, fostering a debate to converge on a final solution.

Automated Multi-Agent Systems: These baselines include the two leading code based MAS generation approaches: **ADAS** and **AFlow**. These methods employ search algorithms to automatically discover the optimal agent roles and their complex interaction topology for a given task domain

Since both ADAS and AFlow are inherently domain-specific and model-specific, we follow the established methodology by training them on individual validation sets to provide the strongest possible baseline performance. ADAS runs in an iterative loop with 30 iterations, following the authors. AFlow runs for until convergence criteria of that method (between 7 to 18 iterations). We use the AFlow code-base provided by the authors as-is. For a fair comparison on mathematical questions, we make a minor modification to the ADAS code-base to disallow Python-based symbolic math tools, instead forcing the model to rely on its inherent reasoning capabilities.

## 4.4 TRAINING

Our training process is designed to independently discover the two core components of our framework: the optimal step-generator module ($m^*$) and the optimal meta-policy ($\pi^*$). This decoupled approach allows us to first forge a powerful, general-purpose reasoning module and then learn a sophisticated strategy to orchestrate it, all without requiring expensive, domain-specific annotations.

**Validation Dataset:** For both discovery processes, we utilize the a subset (1000 samples) of the Math and Science splits of the Open-or-Mixture-of-Thoughts dataset, a general-purpose instruction-following dataset. Our method requires only a one-time, domain-agnostic training phase. The same resulting code artifacts are then deployed across all benchmark domains and foundation models without any task-specific fine-tuning or re-optimization, underscoring the robustness and versatility of our method.

**Step-Generator ($m^*$) Discovery:** We discover the ARM module by employing the Reflection-Guided Evolutionary Search detailed in Algorithm 1. The search is initialized with a basic Chain-of-Thought module ($m_{CoT}$) and iteratively evolves it by maximizing the scaffolded surrogate objective from Section 3.2. This objective evaluates candidate modules within the context of a baseline CoT trace, enabling efficient and stable optimization.

**Meta-Policy ($\pi^*$) Discovery:** The meta-policy is discovered independently using the same evolutionary search algorithm. To ensure computational tractability, this search is performed using the simple and fast baseline module, $m_{CoT}$, as a surrogate for the more complex $m^*$ (as justified in Section 3.3). This allows us to efficiently explore the space of high-level strategies and discover a sophisticated meta-policy that can be seamlessly paired with the optimized ARM module.

## 5 RESULTS

We summarize our results in Table 1 and the key findings are as follows:

**(1) Naive Operators outperform MAS:** Simple basic operators such as CoT, Self-refine, LLM-Debate outperform complex MAS systems like AFlow and ADAS. This highlights an important concern regarding the practicality of recent advancemenets in MAS. On the other hand, simple reasoning operators such as CoT perform substantially better across tasks, and varied families of LLMs. Our ARM based reasoning approach is step forward to revitalize traditional yet strong reasonig methods like CoT, by advancing their reasoning steps with agentic blocks. Our ARM based approach further improves up the CoT performance and achieves best results all the datasets.

**(2) ARM achieving top performance:** ARM consistently outperforms all of the operator baselines. Specifically, in complex datasets such as AIME and HMMT, ARM consistently outperforms existing

| Model | Method | AIME | HMMT | GPQA | LiveBench | Average |
|---|---|---|---|---|---|---|
| GPT-4.1-nano | CoT | 15.1% | 9.9% | 50.0% | 33.1% | 27.0% |
| | CoT-SC | 21.9% | 13.5% | 50.6% | 36.9% | 30.9% |
| | Self-Refine | 17.2% | 9.4% | 50.0% | 28.1% | 26.2% |
| | LLM-Debate | 15.1% | 16.7% | 52.5% | 33.8% | 29.5% |
| | ADAS | 12.0% | 5.2% | 48.1% | 31.2% | 24.1% |
| | AFlow | 18.8% | 12.0% | 39.9% | 30.6% | 25.3% |
| | ARM (**Ours**) | 18.2% | 14.6% | 60.1% | 39.4% | 33.1% |
| | ARM + MP (**Ours**) | **23.4%** | **22.4%** | **61.4%** | **45.6%** | **38.2%** |
| GPT-4o | CoT | 7.3% | 0.5% | 53.8% | 46.2% | 27.0% |
| | CoT-SC | 12.5% | 2.1% | 53.2% | 42.5% | 27.6% |
| | Self-Refine | 6.8% | 2.6% | 53.8% | 37.5% | 25.2% |
| | LLM-Debate | 9.9% | 3.1% | 56.3% | 47.5% | 29.2% |
| | ADAS | 1.0% | 0.0% | 46.2% | 38.8% | 21.5% |
| | AFlow | 9.9% | 3.6% | 53.8% | 41.9% | 27.3% |
| | ARM (**Ours**) | 13.5% | 5.7% | 59.5% | 47.5% | 31.6% |
| | ARM + MP (**Ours**) | **17.2%** | **9.4%** | **60.1%** | **51.9%** | **34.7%** |
| LLaMA-3.3-70B | CoT | 6.8% | 3.1% | 50.0% | 38.1% | 24.5% |
| | CoT-SC | 4.2% | 5.7% | **53.2%** | 45.0% | 27.0% |
| | Self-Refine | 6.8% | 4.2% | 51.3% | 46.9% | 27.3% |
| | LLM-Debate | 5.7% | 4.2% | 50.6% | 46.2% | 26.7% |
| | ADAS | 3.1% | 0.0% | 47.5% | 37.5% | 22.0% |
| | AFlow | 4.7% | 0.0% | 46.8% | 38.1% | 22.4% |
| | ARM (**Ours**) | **8.3%** | 5.2% | 49.6% | 46.2% | 27.3% |
| | ARM + MP (**Ours**) | 7.8% | **6.8%** | 50.0% | **50.0%** | **28.7%** |

Table 1: Main results on four complex reasoning benchmarks across three foundation models. We compare against two groups of baselines: (1) foundational reasoning strategies used to build agentic systems (CoT, CoT-SC, Self-Refine, and LLM-Debate), and (2) existing state-of-the-art automatic MAS design methods (ADAS and AFlow). Our approach is presented in two variants: **ARM**, which recursively applies the discovered reasoning module, and our full method, **ARM + MP**, which combines the ARM with a learned Meta-Policy (MP). Best score in each category is **bolded** and second best score is underlined.

MAS approaches and all the existing baseline operators. This emphasizes the benefits and strong potential of revitalizing proven traditional reasoning methods like CoT.

**(3) Effects from stronger foundation LLM:** We first note an important observation that with stronger LLMs such as GPT-4o, simple operators such as CoT and CoT-SC outperform complex MASes. Our ARM based reasoning approach further pushes the best performance over the baseline operators with both recent stronger frontier models such as GPT4.1-nano / GPT-4o and older benchmark models such as LLaMa-3.3-70B.

# 6 ANALYSES

To understand the sources of ARM's effectiveness, we performed two key analyses. First, we provide empirical evidence that our search objective discovers fundamentally more reliable reasoning modules by minimizing their per-step error rate. Secondly, we show the validity of our efficient, decoupled training strategy by demonstrating that the learned meta-policy transfers zero-shot from a simple surrogate to the final ARM, yielding significant performance gains.

## 6.1 EMPIRICAL VALIDATION OF THE STEP-GENERATOR OBJECTIVE

To empirically validate our theoretical claim (Appendix A) that the scaffolded objective minimizes per-step error, we conducted a targeted ablation study. We executed the top five discovered step-generator modules for a single step, starting from *critical reasoning junctures* identified by an LLM-judge (GPT-OSS-20B) within baseline $m_{CoT}$ traces. The error rate of each single-step output was then evaluated. As shown in Figure 1, a module's rank, determined by our objective, strongly correlates with a lower per-step error rate at these critical points. This result confirms that our search process successfully discovers modules that are fundamentally more robust at a granular level, validating the core mechanism behind ARM's performance.

## 6.2 EMPIRICAL VALIDATION OF META-POLICY TRANSFER

Our methodology relies on a crucial transfer: a meta-policy trained with the simple $m_{CoT}$ module is deployed zero-shot with the powerful, discovered $m^*$ module. The theoretical justification in

| Meta Policy Name (abbreviated) | CoT Baseline | CoT→Meta | Meta Policy |
|---|---|---|---|
| VWASCCoT | 35.1% | 33.7% | 42.0% |
| CWDCWACCCoT | 37.2% | 39.3% | 41.8% |
| RVDCCWASCCoT | 33.7% | 40.0% | 41.8% |
| DRWASCCoT | 35.5% | 34.9% | 41.8% |
| MBECDCCWASCCoT | 36.3% | 39.2% | 41.4% |

Figure 2: Validation of the meta-policy transfer for top discovered policies. The table compares performance using the simple surrogate $m_{CoT}$ (**CoT Baseline**) versus the powerful ARM module $m^*$ (**Meta Policy**). The intermediate **CoT→Meta** column isolates the performance gain from the superior $m^*$ module alone by evaluating it on states generated by the baseline.

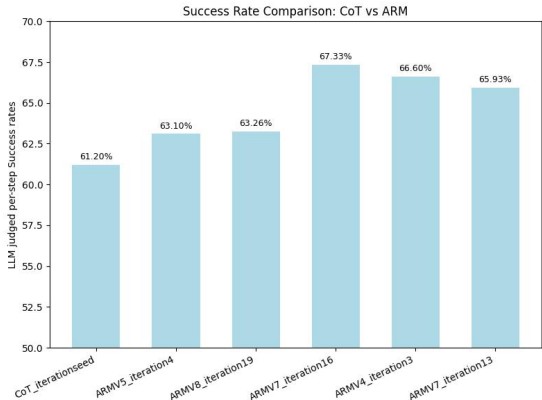

Figure 3: Comparison of LLM judged per-step success rates between the baseline *Chain-of-Thought* (CoT) and multiple *ARM* (CriticChainOfThought) variants. CoT appears first, followed by ARM variants ordered by final performance.

Appendix B posits this transfer is effective due to two factors: (1) the inherent superiority of the $m^*$ module, and (2) its ability to guide the reasoning process into more productive states. We designed an experiment to empirically disentangle and verify these two sources of gain.

To do this, we measure and compare three distinct performance configurations. First, we establish a **baseline performance** using the meta-policy with the simple $m_{CoT}$ module. Second, to isolate the pure **module improvement gain**, we measure the performance of the powerful $m^*$ module when it takes over from intermediate reasoning states generated by the baseline $m_{CoT}$. Finally, we measure the **full system performance** of the meta-policy paired with $m^*$ from the start.

The results, shown in Figure 2, confirm our hypothesis with a clear performance hierarchy. The baseline system performs worst, followed by a significant improvement from simply swapping to the $m^*$ module. The best performance is achieved by the full system, which benefits from both the better module and its ability to find a better reasoning path. This empirically validates the two conditions for successful transfer outlined in AppendixC and confirms the effectiveness of our decoupled discovery strategy.

# 7 CONCLUSION

We introduced ARM, a modular agentic reasoning framework that revitalizes the traditional Chain-of-Thought (CoT) paradigm by augmenting it with lightweight agentic blocks. Through extensive experiments, we demonstrated that simple operators such as CoT and Self-Refine not only remain highly competitive but, in many cases, outperform complex Multi-Agent Systems (MAS), highlighting the growing gap between empirical performance and the perceived promise of increasingly elaborate MAS designs. Our results show that ARM consistently advances the performance of CoT across diverse reasoning tasks and model families, establishing top-performing results.

Beyond empirical improvements, ARM sheds light on an important perspective: progress in reasoning with LLMs may not require increasingly complicated MAS architectures, but rather principled extensions of robust, well-understood methods. By preserving the simplicity and generality of the traditional reasoning method—CoT—while enhancing its reasoning depth and modularity, ARM provides a versatile foundation that can be applied across tasks, domains, and models. Overall, this work underscores the importance of revisiting and strengthening proven reasoning approaches instead of overcomplicating them. ARM represents a step toward a practical, scalable, and broadly applicable modular reasoning approach with LLMs, paving the way for future research to place greater emphasis on simple and generalizable traditional methods for complex reasoning.

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

# A ANALYSES

We conducted two practical analyses to evaluate a) stability of ARM over multiple iterations during search process, b) cost comparison to ARM with prior AMAS techniques.

## A.1 STABILITY OF ARM DURING SEARCH

We evaluated the performance of ARM versions at each iteration to show convergence and stability of the discovered ARM module. As shown in 4, the performance of the ARM modules in later iterations converges to the final performance.

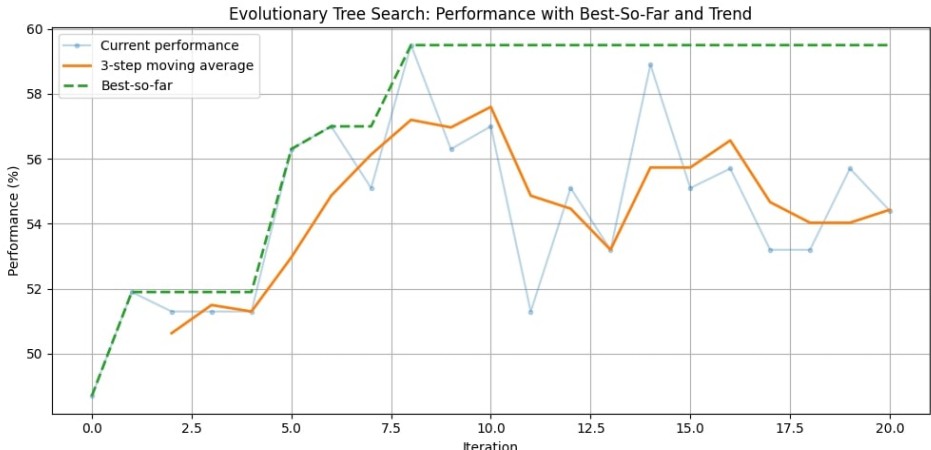

Figure 4: Plot showing performance on GPQA benchmark on intermediate candidates of the search. Note that the search space is discrete (code modules) and the search is tree based exploration/exploitation paradigm, which explains the bumps in performance. However, all intermediate checkpoints are stable, and within 5% of each other.

## A.2 SEARCH AND INFERENCE COST ANALYSIS

There are two costs to analyze: **search cost** and **inference cost**.

| Model | Cost (USD) |
|---|---|
| ARM (Ours) | 4.53 |
| Meta (Ours) | 5.40 |
| **Total Cost (Ours)** | **9.93** |
| ADAS | 2.62 |
| AFlow | 2.52 |

Table 2: Training cost per iteration on 1K samples of open-r1 training data using GPT-4.1-mini, our best performing model

**A) Training Cost per Iteration on 1K Samples (GPT-4.1-mini)**  A single iteration of ARM + Meta policy is $\sim 3.8\times$ the cost of ADAS and $\sim 3.9\times$ the cost of AFlow. However, this must be interpreted correctly:

- Domain- and model-specific techniques such as ADAS, AFlow (and MaAS) must be run **independently for each domain and model**.
- Thus, with 4 domains and 3 models, ADAS cost becomes:
$$4 \times 3 \times 2.62 = \$31.44$$
  and AFlow cost becomes:
$$4 \times 3 \times 2.52 = \$30.24.$$

- In contrast, ARM is **domain- and model-agnostic**, so its amortized cost stays constant—approximately $0.31\times$ that of ADAS and $0.32\times$ that of AFlow.

**Cross-Domain Applicability of ADAS and AFlow**  For fairness, we trained ADAS and AFlow on the same 1K OpenReasoner subset as ARM. Both methods show **poor cross-domain transfer**. Even with model-specific training, we observed no gains when evaluating on unseen domains, indicating that these approaches cannot leverage large-scale datasets beyond the limited benchmark-specific validation sets.

| Method | Model | AIME | HMMT | GPQA | LiveBench |
|---|---|---|---|---|---|
| ADAS (1K) | GPT-4.1-mini | 0.0 | 6.8 | 46.8 | 29.4 |
| ADAS (1K) | GPT-4o | 0.0 | 0.5 | 46.8 | 41.9 |
| ADAS (1K) | Llama-3.3-70B | 3.1 | 0.5 | 42.4 | 46.2 |
| AFlow (1K) | GPT-4.1-mini | 16.7 | 10.4 | 51.3 | 30.6 |
| AFlow (1K) | GPT-4o | 9.4 | 0.0 | 50.6 | 45.0 |
| AFlow (1K) | Llama-3.3-70B | 7.2 | 3.1 | 46.8 | 15.6 |

Table 3: Cross-domain performance when trained on 1K OpenReasoner samples.

Each row above represents a **model-specific but domain-agnostic MAS** configuration, in contrast to ARM which is both **domain- and model-agnostic**.

### A.2.1    B) INFERENCE COST (USD) ON 1K OPENREASONER SAMPLES (GPT-4.1-NANO)

| Method | AIME | HMMT | GPQA | LiveBench |
|---|---|---|---|---|
| ARM (Ours) | 3.69 | 3.02 | 0.81 | 1.12 |
| ARM + Meta (Ours) | 17.60 | 13.77 | 3.22 | 4.36 |
| ADAS | 0.87 | 0.88 | 0.19 | 0.28 |
| AFlow | 0.78 | 0.66 | 0.16 | 0.19 |

Table 4: Inference cost on 1K samples using GPT-4.1-nano.

ARM is designed as a *compute-for-performance* approach.  Instead of running the workflow once—risking severe error propagation—ARM executes the reasoning module at each granular step.  This trades compute for significant performance gains and enables automatic discovery of multi-agentic systems.

**Cost-Optimized ARM Variants.** Since ARM is iteratively refined using a GEPA-style reflection-guided search, adding a cost term to the reviewer agent yields Pareto-optimal cost–performance variants. We will expand on this in the final version of the paper.

| Approach | Cost Multiple | Performance Gain | Gain / Cost |
|---|---|---|---|
| CoT | 1.0 ($0.13) | – | – |
| ADAS | 4.2 | -2.9 | -0.69 |
| AFlow | 3.5 | -1.7 | -0.48 |
| ARM (Ours) | 16.6 | +6.1 | +0.36 |
| ARM + Meta (Ours) | 74.8 | +11.2 | +0.14 |

Table 5: Cost vs. performance comparison.

**Cost–Performance Comparison with CoT and MAS Approaches**  ADAS, AFlow, and other MAS methods are substantially more expensive than simple CoT yet offer only marginal or negative gains. ARM, while costlier, delivers **significantly higher and consistent performance**, with a more favorable cost-normalized improvement ratio compared to all other automated agentic systems.

## B  ARM SEARCH ALGORITHM

Algorithm 1 provides the full pseudocode of the reflection-guided search algorithm for evolving ARM modules.

---

**Algorithm 1** Reflection-Guided Search

---

1: **Input:** Initial program $p_{root}$ (e.g., $m_{CoT}$ or $\pi_{Rec}$), evaluation function EVALUATE($\cdot$), total iterations $K$, exploration constant $C$.
2: **Initialize:**
3: Tree $\mathcal{T}$ with a single node for $p_{root}$.
4: $p_{root}.\bar{\mathcal{R}} \leftarrow$ EVALUATE($p_{root}$)           ▷ Evaluate the baseline program on a validation batch
5: $p_{root}.N \leftarrow 1$                                  ▷ Initialize visit count for the root
6: **for** $t = 1$ **to** $K$ **do**
7:                                        ▷ *1. Select a parent program to mutate*
8:      $P(p_i) \leftarrow \frac{\exp(p_i.\bar{\mathcal{R}}/T)}{\sum_{j \in \mathcal{T}} \exp(p_j.\bar{\mathcal{R}}/T)}$
9:      $p_{parent} \leftarrow$ Sample($\mathcal{T}, P$)
10:                                    ▷ *2. Expand the tree via reflection*
11:      traces $\leftarrow$ EXECUTE($p_{parent}$)                ▷ Collect execution traces
12:      history $\leftarrow$ GETMUTATIONHISTORY($p_{parent}$)
13:      $p_{new} \leftarrow$ REVIEWERAGENT($p_{parent}$, traces, history)
14:                                    ▷ *3. Evaluate the new program*
15:      $p_{new}.\bar{\mathcal{R}} \leftarrow$ EVALUATE($p_{new}$)
16:      $p_{new}.N \leftarrow 1$
17:                                    ▷ *4. Update tree and statistics*
18:      $\mathcal{T}$.ADDCHILD($p_{parent}, p_{new}$)
19:      $p_{parent}.N \leftarrow p_{parent}.N + 1$
20: **end for**
21:
22: **return** $\underset{p_i \in \mathcal{T}}{\text{argmax}} \ (p_i.\bar{\mathcal{R}})$          ▷ Return the program with the highest empirical reward

---

## C  THEORETICAL ANALYSIS

A complete theoretical analysis of the multi-agentic system ARM powered by LLMs is intractable due to the complex, high-dimensional nature of language generation and the non-stationary of the generation process. Recent research (Chang et al., 2025; Kim et al., 2025) models sequential CoT reasoning steps as a Markov Decision Process by abstracting away the underlying complexities of the text generation process and focusing on higher level reasoning states. Therefore, to build a formal intuition for the design choices in our scaffolded search for the step-generator, and the decoupled search for the meta-policy (Algorithm1), we also analyze an idealized formulation of the problem as a Markov Decision Process (MDP).

Our analysis is particularly inspired by recent work on self taught reasoners (RL-Star) by Chang et al. (2025), where they introduce a step indexed competence parameter $\delta_{t,n}$ which quantifies the advantage in probability of a correct reasoning step at step $n$ during training iteration $t$ over a baseline random reasoner. They show the conditions under which a bootstrapped RL learning algorithm based on rejection sampling shows monotonic improvement and convergence. While our goals are similar (improving the reasoning process), our problem statement has critical differences which makes a straight forward adaption infeasible: RL-Star analyses a system where the LLM's parametric weights are updated via reinforcement learning. On the other hand, ARM treats the LLM as a black box and performs discrete, evolutionary search (Fernando et al., 2024; Agrawal et al., 2025) over programming modules that orchestrate calls to the LLM. Consequently, our search is inherently *discrete*, so smoothness-based guarantees do not apply. Hence we do not assume or prove convergence guarantees, and instead motivate the intuition of our scaffolded search process as a conservative policy improvement (CPI) (Kakade & Langford, 2002) that preferentially selects modules with higher competence leading to improved reasoning process.

## C.1  AN IDEALIZED MDP MODEL OF STEP-WISE REASONING

We model the reasoning process as a Markov decision process (MDP) Sutton & Barto (2018) $\mathcal{M} = (S, N, A, P, R, \gamma)$:

- **State Space** $(S)$**:** The state space $S = \mathcal{U} \cup \mathcal{G} \cup \mathcal{F}$ is partitioned into three disjoint subsets:
  - $\mathcal{U}$: A state $s \in \mathcal{U}$ represents a partial reasoning trace $q, p_1, ...p_k$ that is not yet terminated.
  - $\mathcal{G}$: A state $s \in \mathcal{G}$ represents a reasoning path that has successfully ended on the right answer. In our setting this is when the module emits the `/boxed{correct answer}`. This is an absorbing region.
  - $\mathcal{F}$: A state $s \in \mathcal{F}$ represents a reasoning path that has terminated at the wrong answer. In our setting this is when the module emits the `/boxed{incorrect answer}`. This is an absorbing region.

- **Verification Predicate (`solved`):** A predicate function $\mathcal{S} \to 0, 1$ judging if the right answer is already derivable from the given partial reasoning state. Note that this is a simple formatting action, and is independent of the module $m$.
  - `solved(s) = 0` $\forall s \in \mathcal{F}$
  - `solved(s) = 1` $\forall s \in \mathcal{G}$

- **Maximum Reasoning Steps (N)**: We rollout the reasoning process up to $N$ steps. After $N$ steps of reasoning, we enforce a *model-independent* termination rule where the state deterministically goes into $s' \in \mathcal{G}$ if `solved(s) = 1` and into $s' \in \mathcal{F}$ if `solved(s) = 0`. For simplicity of notation, we assume the total trajectory length to be $N + 1$.

- **Action Space** $(A)$**:** For a fixed meta-policy $\pi_{Rec}$ that recursively generates steps until termination (such as the one used by baseline CoT or the ARM-only variant), the meta policy executes a single action at any give state $s \in \mathcal{U}$: i.e., invokes a step-generator module $m$ to produce the next reasoning step. Thus, the action space is a singleton $\mathcal{A} = \{$`generate_step`$\}$. For terminal states $\mathcal{G} \cup \mathcal{F}$, this is a `no-op`. Hence, the choice of the module $m$ fully defines the transition dynamics of the MDP.

- **Reward Function** $(R)$**:** The one-shot terminal reward is sparse:

$$R(s \to s') = \begin{cases} 1, & s' \in \mathcal{G}, \\ 0, & \text{otherwise.} \end{cases}$$

- **Transition Dynamics (P)**: We denote the state transition probability $P(.|s, m)$ with the Markov assumption. This simplification is the core foundation for our MDP analysis.

- **Value Function:** For $n \in \{0, \ldots, N\}$, let $V_n^m(s)$ denote the value with $n$ reasoning steps remaining before the formatting step. The Bellman recursion can be written as

$$V_n^m(s) = \begin{cases} 1, & s \in \mathcal{G}, \\ 0, & s \in \mathcal{F}, \\ \{\texttt{solved}(s)\} + \{\neg\texttt{solved}(s)\} \, \mathbb{E}_{s' \sim P_m(\cdot|s)}[V_{n-1}^m(s')], & s \in \mathcal{U}, n \geq 1, \\ \{\texttt{solved}(s)\}, & s \in \mathcal{U}, n = 0. \end{cases}$$

Within this MDP framework, the ideal objective is to discover a module $m^*$, that maximizes the expected value from the initial state distribution $d_0(s)$

$$m^* = \arg \max_{m \in \mathcal{M}} \mathbb{E}_{s_0 \sim d_0(s)} [V_N^m(s_0)]$$

This objective poses several major optimization challenges: 1) credit assignment problem over long sequence of steps and 2) unconstrained search space of code modules.

## C.2 DEFINITIONS

We introduce the following quantities to characterize module's performance and search strategy:

- **Per-step competence** $\delta_m(s)$**:** This represents the *competence* of the module $m$ at a reasoning state $s \in \mathcal{U}$ analogous to $\delta_{t,n}$ term in Chang et al. (2025). The probability that a one step update $s \in \mathcal{U}$ is valid can be viewed as a monotonically increasing function over $\delta_m(.)$, but for simplicity of notation, we assume this to be $\delta_m(s)$ itself.

- **Recovery** $r_m(s)$**:** The probability that, conditioned on an invalid one-step update from $s$, the next step returns to a valid state. This term captures the recovery possibility of a mistake in the immediate next turn. While recovery can happen at any turn following the mistake in a real LLM, we limit the window to 1 for simplicity.

- **Composite Validity** $\phi_m(s)$**:** The total probability that the next step is valid for a step $s \in \mathcal{U}$, either by being immediately valid, or by being successfully repaired on the next step:

$$\phi_m(s) = \delta_m(s) + (1 - \delta_m(s))r_m(s) \tag{1}$$

- **Window** $W = (n, l)$**:** A block of $l$ consecutive steps starting at step index $n$ where all states remain in $\mathcal{U}$. The ARM module replaces the baseline module $m_{CoT}$ with a candidate module $m$ only on this window.

- **Visitation Weights** $w^\pi(W)$**:** The probability under baseline policy $(\pi, m_{CoT})$ that the window $W$ occurs. This measures the frequency with which the meta-policy starts the module at a given window.

## C.3 KEY SIMPLIFYING ASSUMPTIONS

The rest of our analysis relies on the following key assumptions.

**Assumption 1.** [Local competence lift in the scaffolding window] Within any given window $W = (n, l)$, for all states $s$ visited, the candidate model satisfies $\phi_m(s) \geq \phi_{m_{CoT}(s)} + \Delta_c$ for some lift $\Delta_c \in [0, 1)$.

*Rationale:* This is the empirical premise that our scaffolded search objective (Section 3.2) is designed to optimize for. Our Algorithm 1 directly measures and selects for modules that improve local validity and recovery rates over the CoT within a constrained context at random locations.

**Assumption 2.** [Compatibility Loss] We define $\beta_l(W)$ a bound on the probability that replacing the baseline $m_{CoT}$ with ARM module $m$ at a window $W$ yields a context which is unusable for the rest of the baseline reasoning trace. We refer to $(1 - \beta_l(W))$ as the *compatibility factor*. Furthermore, we define $\overline{\beta}_l \sup_{W \in \mathcal{W}_{\text{valid}}} \beta_l(W)$ as the supremum of incompatibility probabilities across all valid windows, representing the worst-case incompatibility bound.

*Rationale:* Swapping the baseline module $m_{CoT}$ with $m$ can introduce a "*context drift*" or "*semantic drift*" which could amplify at deployment time when the ARM module $m$ is used through the entire trajectory. Our approach minimizes this drift by two means: 1) the few shot examples of the progress, as well as the provided partial progress acts as a powerful inductive bias to constrain the next step to states that preserve usefulness, for example by adopting the same notation, logical continuity, etc. 2) the reviewer agent which proposes mutations to the module (starting from baseline CoT) is prompted to generate modules which solve one step at a time starting from the given partial progress.

**Assumption 3.** [Module-invariant termination] We assume that the reward is terminal, and is provided under the condition that the extracted answer matches the right answer. Furthermore, the last step in the MDP is reserved for this extraction, which is considered module-invariant, i.e. both CoT or any other module can do this final syntax step (equally) perfectly.

C.4 THEORETICAL GROUNDING FOR THE SCAFFOLDED STEP-GENERATOR SEARCH

The scaffolded objective evaluates a candidate $m$ by *splicing* it into a baseline rollout for a short window $t \in \{i, \ldots, i + \ell - 1\}$ while keeping $m_{\text{CoT}}$ before and after:

$$\underbrace{U^*_{m_{\text{CoT}}} \circ \left( U^\ell_m \right) \circ U^i_{m_{\text{CoT}}}}_{\text{"baseline–candidate–baseline"}}.$$

This section formalizes the link between local module improvements and global performance gains. Under our simplified MDP framework and assumptions 1, 2, we establish the following lemmas:

**Lemma 1.** [Window lift from local competence] The probability of remaining in $\mathcal{U}$ after the window increases by at least $l.C^{l-1}.\Delta_c$ for some constant $C \in (0, 1)$.

*Proof:* The $l$ step window survival (being in $\mathcal{U}$) is at least $\phi_m(s)^l$. Under Assumption 1, the per-step composite validity improves by at least $\Delta_c$. Hence, the survival probability improves by at least $\left( \phi_{\text{CoT}}(s) + \Delta_c \right)^l - \phi_{\text{CoT}}(s)^l$. Applying the mean value theorem on $f(x) = x^\ell$, we get

$$(x + \Delta_c)^\ell - x^\ell = \ell \xi^{\ell-1} \Delta_c \geq \ell (\min_{s \in \mathcal{U}_\mathcal{R}} \phi_{\text{CoT}}(s))^{\ell-1} \Delta$$

for some $\xi \in [x, x + \Delta]$, where $\mathcal{U}_R$ represents states reachable by baseline $CoT$ module. Further, let $C := \min_{s \in U_R} \phi_{\text{CoT}}(s)$ be a constant.

**Lemma 2.** [Accounting for Compatibility] The probability of a sample surviving the window $W$ and remain usable after the module swap is lower bounded by $(1 - \beta_l(W)).(l.C^{l-1}.\Delta_c)$ where C is the constant from Lemma 1.

*Proof:* From Assumption 2, the usability probability is at least $(1 - \beta_l(W))$. Multiplying this by the probability of surviving the window from Lemma 1 yields the result.

**Lemma 3.** [From window survival to finite-horizon success] Any increase in the probability of staying within $\mathcal{U}$ across a window $W$ (while remaining usable) weakly increases the probability of reaching $\mathcal{G}$ within the horizon $N$.

*Proof:* Under assumption 3, the termination rule is module-invariant, and reaching the goal state only depends on being in state $s \in \mathcal{U} \cup \mathcal{G}$ and `solved(s)=1`. Thus a higher probability of preserving valid, in-progress states across window cannot decrease (and generally increases) the likelihood that subsequent steps will generate such a solvable state before the horizon is exhausted. This follows from standard monotonicity arguments on absorbing Markov chains.

**Theorem 1** (Gain from Scaffolded Module Substitution in recursive meta policy). Let $J(\pi_{Rec}, m) = \mathbb{E}\left[V^{\pi_{Rec}, m}_N(s_0)\right]$ denote the expected terminal reward (success probability) obtained when recursively applying the step-generator module $m$ under a fixed baseline meta-policy $\pi_{Rec}$ for horizon $N$. The improvement of the ARM module $m^*$ over $m_{CoT}$ is at least:

$$J(\pi_{Rec}, m^*) - J(\pi_{Rec}, m_{\text{CoT}}) \geq \sum_W w_\pi(W) \kappa(W) (1 - \beta_l(W)) l C^{l-1} \Delta_c. \tag{2}$$

where each term represents:

- $w_\pi(W)$: visitation probability of a window $W$ under a baseline rollout;

- $\kappa(W)$: probability that a usable post-window state leads to terminal success within the remaining horizon.

- $C \in (0, 1]$: a constant from Lemma 1 capturing compounding survival over steps.

In particular, if $\kappa(W) \geq \kappa_{min} \geq 0$ for all $W$, then

$$J(\pi_{Rec}, m^*) - J(\pi_{Rec}, m_{\text{CoT}}) \geq \kappa_{min} (1 - \beta_l(W)) l C^{l-1} \sum_W w_\pi(W) \tag{3}$$

*Proof:* From Lemma 2 , the probability of surviving the window is lower bounded by $(1 - \beta_l) l C^{l-1}$. Let $\kappa(W)$ represent the probability of success upon starting from a good state, post the window. By

Lemma 3, the increase in usable post-window mass translates to atleast a $\kappa(W)$ fraction improvement in terminal success within the remaining horizon. Thus the expected gain from the window is $\kappa(W)(1 - \beta_l(W))lC^{l-1}$. Taking expectation over window visitation probabilities yields the result:

$$
\begin{aligned}
J(\pi_{Rec}, m^*) - J(\pi_{Rec}, m_{\text{CoT}}) &= \sum_W w_\pi(W) \cdot \text{Gain}(W) \\
&= \sum_W w_\pi(W) \cdot \kappa(W) \left(1 - \beta_l(W)\right) l \, C^{l-1}. \\
&\geq \sum_W w_\pi(W) \cdot \kappa_{min} \left(1 - \beta_l(W)\right) l \, C^{l-1}.
\end{aligned}
\tag{4}
$$

where $k_{min} \geq 0$ is the lowest probability of success from a valid, usable intermediate reasoning trace.

## C.5 THEORETICAL JUSTIFICATION FOR ZERO-SHOT POLICY TRANSFER

The learned meta policy $\pi^*$ uses $m_{CoT}$ as the *step generator* during the learning phase and is deployed zero-shot using the discovered ARM module $m^*$. Below, we justify why this transfer is effective.

**Theorem 2** (Validity of Zero-shot step generator swap in Meta policy). Let $J(\pi^*, m^*) = \mathbb{E}_{s_0 \sim \mathcal{D}} \left[ V_N^{(\pi^*, m^*)}(s_0) \right]$ denote the expected terminal reward obtained when applying the discovered meta-policy $\pi^*$ (from Section 3.3), with the step-generator module $m^*$ under horizon $N$. If $\Delta_c \geq \frac{\overline{\beta_l}}{1 - \overline{\beta_l}}$, then the transfer is valid, i.e., $J(\pi^*, m^*) \geq J(\pi^*, m_{CoT})$

Let's define per-step advantage of a module $m$ over $m_{CoT}$ with $n$ more steps to go as the expected difference in value when taking one step with $m$ and the rest with $m_{CoT}$:

$$
A_n(s_n, m) \triangleq \mathbb{E}_{s' \sim P_m(\cdot|s)}[V_{n-1}^{(\pi^*, m_{CoT})}(s')] - \mathbb{E}_{s' \sim P_{m_{CoT}}(\cdot|s)}[V_{n-1}^{(\pi^*, m_{CoT})}(s')]
\tag{5}
$$

Now let's consider the difference in expected value starting from a given state $s_0$ sampled from the data distribution $\mathcal{D}$. For simplicity, we drop $\pi^*$ from notation as it is the common meta policy in both terms.

$$
V_N^m(s_0) - V_N^{m_{CoT}}(s_0)
$$

Rolling out for one step yields

$$
\mathbb{E}_{s_1 \sim P_m(\cdot|s_0)}[V_{n-1}^m(s_1)] - V_n^{m_{CoT}}(s_0)
$$

Adding an subtracting $\mathbb{E}_{s_1 \sim P_m(\cdot|s_0)}[V_{n-1}^{m_{CoT}}(s_1)]$ (i.e., sampling from $m$ but continue with $m_{CoT}$) we get:

$$
\begin{aligned}
\mathbb{E}_{s_1 \sim P_m(\cdot|s_0)}[V_{n-1}^{m_{CoT}}(s_1)] - \mathbb{E}_{s_1 \sim P_{m_{CoT}}(\cdot|s_0)}[V_{n-1}^{m_{CoT}}(s_1)] \\
+ \mathbb{E}_{s_1 \sim P(\cdot|s_0)}[V_{n-1}^m(s_1) - V_{n-1}^{m_{CoT}}(s_1)]
\end{aligned}
\tag{6}
$$

By Equation-5, this can be written as

$$
A_n^{m_{CoT}}(s_0, m) + \mathbb{E}_{s_1 \sim P(\cdot|s_0)}[V_{n-1}^m(s_1) - V_{n-1}^{m_{CoT}}(s_1)]
$$

This is a recursive equation in $n$ since the second term is the difference in value between the module with $n - 1$ steps to horizon. Hence:

$$
V_N^m(s_0) - V_N^{m_{CoT}}(s_0) = \sum_{n=0}^N \left[ A_{N-n}^{m_{CoT}}(s_n, m) \right]
\tag{7}
$$

Thus we can conservatively guarantee module improvement, if each of the the advantage term is positive. Suppose that $U$ represents the event that one step rollout using our discovered module $m^*$ is usable (i.e. no errors, and usable context) in the next turn, then by law of total expectation the advantage term can be written as:

$$\mathbb{P}(U \mid s, m^*) \cdot \big(\mathbb{E}[V_{N-n} \mid s, m^*, U] - \mathbb{E}[V_{N-n} \mid s, m_{\text{CoT}}, U]\big)$$
$$+ \mathbb{P}(\neg U \mid s, m^*) \cdot \big(\mathbb{E}[V_{N-n} \mid s, m^*, \neg U] - \mathbb{E}[V_{N-n} \mid s, m_{\text{CoT}}, \neg U]\big) \tag{8}$$

By Assumption 1 and Assumption 2, the first term is at least $(1 - \overline{\beta_l}) \cdot \Delta_c$. The second term is lower bounded in the worst case by $\overline{\beta_l} \cdot (-1)$ since the probability of non-useful state is $\overline{\beta_l}$ and the difference in reward is at most $-1$ (when $m^*$ fails where $m_{CoT}$ succeeds). Thus a conservative lower bound is:

$$A_n^{m_{CoT}}(s, m) \geq (1 - \overline{\beta_l}) \cdot \Delta_c - \overline{\beta_l} \tag{9}$$

Hence advantage is non-negative when $\Delta_c \geq \frac{\overline{\beta_l}}{1 - \overline{\beta_l}}$, and we can guarantee that zero-shot transfer is effective when this condition is satisfied.

**Corrolary 1.** The meta policy, paired with ARM, $(\pi^*, m^*)$ yields the highest performance when $\Delta_c$ is large and $\beta_l$ is small.

*Remark:* From Equation-8, the claim holds. Our scaffolded objective function is precisely designed to maximize $\Delta_c$ while reducing the probability of state drift leading to mismatched compatibility. In 3, we can empirically see that better performance is associated with improvement in per-step competence. And in 2, we see that the ARM module is compatible with $m_{CoT}$ and can act as a superior drop-in replacement for CoT. Furthermore, the examples shown in Section D show that both CoT and ARM rollouts are empirically non-discriminable.

## D  STEP GENERATOR ROLL OUT EXAMPLES

The scaffolded objective function (Section 3.2) deliberately ensures that the discovered ARM module ($m^*$) is a strict drop-in **refinement** of the CoT operator rather than a qualitatively new operator. This ensures that compute scaling is performed at granular reasoning level, while learning interpretable, CoT like behaviors while reducing error propagation (Section 6.1& Section C). As seen in the following examples, the final rollout output of the ARM module closely resembles that of the CoT operator.

### D.1  EXAMPLE-1

**ARM Rollout**

```
### Step 2
Express the sides in terms of the common ratio \(2R\):
\[
a = 2R \sin 60^\circ = 2R \times \frac{\sqrt{3}}{2} = R \sqrt{3},
\]
\[
b = 2R \sin 45^\circ = 2R \times \frac{\sqrt{2}}{2} = R \sqrt{2},
\]
\[
c = 2R \sin 75^\circ.
\]
Recall that \(\sin 75^\circ = \sin(45^\circ + 30^\circ)\):
\[
\sin 75^\circ = \sin 45^\circ \cos 30^\circ + \cos 45^\circ \sin 30^\
    circ = \frac{\sqrt{2}}{2} \times \frac{\sqrt{3}}{2} + \frac{\sqrt
    {2}}{2} \times \frac{1}{2} = \frac{\sqrt{6}}{4} + \frac{\sqrt
    {2}}{4} = \frac{\sqrt{6} + \sqrt{2}}{4}.
\]
Hence,
\[
c = 2R \times \frac{\sqrt{6} + \sqrt{2}}{4} = R \times \frac{\sqrt{6}
    + \sqrt{2}}{2}.
\]
```

**CoT Rollout**

```
### Step 2
Expressing the sides in terms of the circumradius \( R \):
\[
a = 2R \sin A = 2R \sin 60^\circ = 2R \times \frac{\sqrt{3}}{2} = R \
    sqrt{3},
\]
\[
b = 2R \sin B = 2R \times \frac{\sqrt{2}}{2} = R \sqrt{2},
\]
\[
c = 2R \sin C = 2R \times \sin 75^\circ.
\]
Recall that:
\[
\sin 75^\circ = \sin (45^\circ + 30^\circ) = \sin 45^\circ \cos 30^\
    circ + \cos 45^\circ \sin 30^\circ = \frac{\sqrt{2}}{2} \times \
    frac{\sqrt{3}}{2} + \frac{\sqrt{2}}{2} \times \frac{1}{2} = \frac
    {\sqrt{6}}{4} + \frac{\sqrt{2}}{4} = \frac{\sqrt{6} + \sqrt
    {2}}{4}.
\]
Therefore:
\[
c = 2R \times \frac{\sqrt{6} + \sqrt{2}}{4} = R \times \frac{\sqrt{6}
    + \sqrt{2}}{2}.
\]
```

**ARM Rollout**

```
### Step 3
Now, determine the new temperature after doubling the kinetic energy:
\[
T_2 = 526.3\,K
\]
Convert this back to Celsius:
\[
T_2^{\circ} C = 526.3 - 273.15 = 253.15^\circ C
\]
This is a significant increase from the initial \(-10^\circ C\),
    indicating the molecules now have enough energy corresponding to
    about \(253^\circ C\).
```

**CoT Rollout**

```
### Step 3
Determine the new temperature after doubling the kinetic energy:
Since kinetic energy \( \propto T \),
\[
T_2 = 2 \times T_1 = 526.3 \, K
\]
Convert this back to Celsius:
\[
T_2 (^\circ C) = T_2 (K) - 273.15 = 526.3 - 273.15 \approx 253.15^\
    circ C
\]
This temperature is well above the boiling point of water at 1 atm
    (100C). Therefore, the molecules' kinetic energy would be
    sufficient to overcome intermolecular forces and convert the water
     from the solid phase (ice) into the gas phase (vapor).
```

## D.3 EXAMPLE-3

---

**ARM Rollout**

```
### Step 1

To determine the correct substitution from equation 12.9 to equation
    12.10, we need to understand the context of these equations and
    the variables involved. Equation 12.9 likely involves a momentum-
    like variable \( P^+ \) or a gamma matrix that acts on spinors,
    and the goal is to simplify or rewrite this expression into a form
     involving gamma matrices or energy-momentum components.

Typically, in relativistic quantum mechanics or quantum field theory,
    the transformation from the light-cone momentum \( P^+ \) to gamma
     matrices involves choosing a specific gamma matrix to project or
    decompose spinor components. The options provided suggest that the
     substitution involves gamma matrices and possibly the momentum
    operator or gamma matrices related to time.

Given that, the known structures are:

- \( P^+ \): light-cone momentum component, usually \( P^+ = P^0 + P^3
     \).
- \( \gamma^u \): a gamma matrix with a specific index \( u \),
    possibly a spatial index.
- \( \gamma^0 \): the gamma matrix corresponding to time.
- \( \gamma^0 P \): a product of \( \gamma^0 \) with some momentum
    operator \( P \).

Since the transformation is from equation 12.9 to 12.10, and the
    options involve gamma matrices and momenta, the typical approach
    in such derivations involves rewriting the projection operator or
    the momentum in terms of gamma matrices.
```

```
CoT

### Step 1

To determine the correct substitution, first understand the context:
    equation 12.9 involves the momentum operator \( P^\mu \), and
    equation 12.10 is likely a transformed version involving gamma
    matrices. In relativistic quantum mechanics, transformations often
     involve gamma matrices \( \gamma^\mu \) and the gamma matrices'
    role in formulating Dirac equations.

Given the options:

- A: \( P^+ \)
- B: \( \gamma^u \)
- C: \( \gamma^0 \)
- D: \( \gamma^0 P \)

and based on the typical transformations, the substitution from \( P^\
    mu \) to a form involving gamma matrices usually involves \( \
    gamma^0 \) because it relates to the time component of the gamma
    matrices and appears frequently in the Dirac formalism.

In particular, if the original equation involves \( P^\mu \), and the
    transformed equation involves a gamma matrix, then the key is
    recognizing the common substitution that converts momentum
    operators into gamma matrix expressions, especially the time
    component \( P^0 \).

Therefore, the step is to analyze which gamma matrix substitution
    correctly transforms the operator form in equation 12.9 into the
    one in 12.10.
```

## E  BEST ARM DISCOVERED: CRITICCHAINOFTHOUGHTV7

The following is the Python implementation of the best ARM discovered by our algorithm.

```python
import asyncio

class CriticChainOfThoughtV7:
    def __init__(self, llm):
        self.llm = llm

    async def forward(self, problem, partial_progress):
        # 1. Generate four candidate next steps in parallel
        candidate_tasks = [
            self.llm.generate_step(problem, partial_progress)
            for _ in range(4)
        ]
        candidates = await asyncio.gather(*candidate_tasks)

        # 2. Critique candidates in two groups of two, in parallel
        critique_tasks = []
        groups = [
            (0, 2, ("rating_1", "rating_2"), ("critique_1",
            "critique_2")),
            (2, 4, ("rating_3", "rating_4"), ("critique_3",
            "critique_4"))
        ]
        for start, end, rating_names, critique_names in groups:
            context = [
                {
                    "name": "Problem",
                    "data": problem,
                    "description": "The problem to solve."
                },
                {
                    "name": "Partial Progress",
                    "data": partial_progress,
                    "description": "The partial solution so far."
                },
                {
                    "name": "Candidate Next Steps",
                    "data": "\n\n".join(
                        f"### Candidate Next Step
                        {i+1}\n{candidates[i]}"
                        for i in range(start, end)
                    ),
                    "description": "Two candidate next steps
                    formatted with markdown subheaders."
                }
            ]
            instructions = (
                "You are given a problem, the current partial
                solution, and two candidate next reasoning steps.\n"
                "For each candidate, provide:\n"
                f"- {rating_names[0]} and {rating_names[1]}: a single
                integer rating from 1 to 10 indicating its fit as the next
                reasoning step (10 is best).\n"
                f"- {critique_names[0]} and {critique_names[1]}: a
                one-sentence critique highlighting each candidate's strengths
                and weaknesses.\n"
                f"Name the fields exactly {rating_names[0]},
                {critique_names[0]}, {rating_names[1]}, {critique_names[1]}."
            )
            response_format = [
                {
```

```python
                        "name": rating_names[0],
                        "description": f"Integer rating (1-10) for
    ↪ Candidate Next Step {start+1}."
                    },
                    {
                        "name": critique_names[0],
                        "description": f"One-sentence critique of
    ↪ Candidate Next Step {start+1}."
                    },
                    {
                        "name": rating_names[1],
                        "description": f"Integer rating (1-10) for
    ↪ Candidate Next Step {start+2}."
                    },
                    {
                        "name": critique_names[1],
                        "description": f"One-sentence critique of
    ↪ Candidate Next Step {start+2}."
                    }
                ]
            critique_tasks.append(
                self.llm.chat_completion(context, instructions,
    ↪ response_format)
            )

        critiques = await asyncio.gather(*critique_tasks)

        # 3. Parse ratings and identify the two highest-rated
    ↪ candidates
        ratings = [
            int(critiques[0]["rating_1"]),
            int(critiques[0]["rating_2"]),
            int(critiques[1]["rating_3"]),
            int(critiques[1]["rating_4"])
        ]
        sorted_indices = sorted(range(4), key=lambda i: ratings[i],
    ↪ reverse=True)
        top1_idx, top2_idx = sorted_indices[0], sorted_indices[1]
        top1_candidate = candidates[top1_idx]
        top2_candidate = candidates[top2_idx]

        # 4. Final head-to-head comparison between the top two
    ↪ candidates
        context_final = [
            {
                "name": "Problem",
                "data": problem,
                "description": "The problem to solve."
            },
            {
                "name": "Partial Progress",
                "data": partial_progress,
                "description": "The partial solution so far."
            },
            {
                "name": "Candidate Next Steps",
                "data": (
                    f"### Candidate A\n{top1_candidate}\n\n"
                    f"### Candidate B\n{top2_candidate}"
                ),
                "description": "Two top candidate next steps
    ↪ formatted with markdown subheaders."
            }
        ]
        instructions_final = (
```

```python
107            "Compare Candidate A and Candidate B as the next
       ↪ reasoning step for the given problem and partial progress.\n"
108            "Provide:\n"
109            "- winner: choose either 'Candidate A' or 'Candidate B'
       ↪ indicating which step is better.\n"
110            "- justification: one-sentence justification for your
       ↪ choice."
111        )
112        response_format_final = [
113            {
114                "name": "winner",
115                "description": "Either 'Candidate A' or 'Candidate B'
       ↪ indicating the better next step."
116            },
117            {
118                "name": "justification",
119                "description": "One-sentence justification for the
       ↪ choice."
120            }
121        ]
122        final_decision = await self.llm.chat_completion(
123            context_final, instructions_final, response_format_final
124        )
125
126        if final_decision["winner"].strip() == "Candidate A":
127            selected_candidate = top1_candidate
128            runnerup_candidate = top2_candidate
129        else:
130            selected_candidate = top2_candidate
131            runnerup_candidate = top1_candidate
132
133        # 5. Post-selection adversarial critique with severity rating
134        context_flaw = [
135            {
136                "name": "Problem",
137                "data": problem,
138                "description": "The problem to solve."
139            },
140            {
141                "name": "Partial Progress",
142                "data": partial_progress,
143                "description": "The partial solution so far."
144            },
145            {
146                "name": "Selected Candidate Next Step",
147                "data": f"### Selected Candidate Next
       ↪ Step\n{selected_candidate}",
148                "description": "The final chosen candidate next
       ↪ reasoning step formatted with a markdown subheader."
149            }
150        ]
151        instructions_flaw = (
152            "You are given a problem, the current partial solution,
       ↪ and a selected next reasoning step.\n"
153            "Identify any major flaw or missing piece of reasoning in
       ↪ the selected step.\n"
154            "Provide:\n"
155            "- flaw: either the single word 'None' if there is no
       ↪ flaw, or a brief description of the flaw.\n"
156            "- severity: a single integer rating from 1 to 10
       ↪ indicating how severe the flaw is (10 is critical)."
157        )
158        response_format_flaw = [
159            {
160                "name": "flaw",
```

```
161                "description": "Either the single word 'None' if
    ↪ there is no flaw, or a brief description of a major flaw in
    ↪ the selected step."
162            },
163            {
164                "name": "severity",
165                "description": "Integer rating (1-10) indicating
    ↪ severity of the flaw (10 is most severe)."
166            }
167        ]
168        flaw_response = await self.llm.chat_completion(
169            context_flaw, instructions_flaw, response_format_flaw
170        )
171        flaw = flaw_response["flaw"].strip()
172        severity = int(flaw_response["severity"])
173
174        # 6. Compute dynamic severity threshold based on rating gap
175        gap = ratings[top1_idx] - ratings[top2_idx]
176        if gap <= 1:
177            threshold = 5
178        elif gap == 2:
179            threshold = 6
180        else:
181            threshold = 7
182
183        # 7. If a severe flaw is detected above the dynamic
    ↪ threshold, fall back
184        if flaw.lower() != "none" and severity >= threshold:
185            return runnerup_candidate
186        return selected_candidate
```

Listing 1: Code for CriticChainOfThoughtV7, performance: 38.0

## F BEST META-POLICY DISCOVERED: VERIFIEDWEIGHTEDADAPTIVESELFCONSISTENTCHAINOFTHOUGHT

The following is the Python implementation of the best meta-policy discovered by our algorithm.

```python
1  import asyncio
2  from agent.solution import Solution, Step
3  from judge_utils import judge_equality
4
5  class VerifiedWeightedAdaptiveSelfConsistentChainOfThought:
6      def __init__(self, llm, block):
7          self.llm = llm
8          self.block = block
9
10     async def forward(self, problem):
11         # Helper: generate one chain up to 8 steps, then complete via
    ↪ LLM if needed
12         async def generate_chain():
13             solution = Solution()
14             for _ in range(8):
15                 next_step = await self.block.forward(problem,
    ↪ str(solution))
16                 solution.add_step(Step(str(next_step)))
17                 if solution.is_completed():
18                     return solution
19             completion = await self.llm.complete_solution(problem,
    ↪ str(solution))
20             solution.add_step(Step(str(completion)))
21             return solution
```

```python
          # Helper: confidence scoring (1-5)
          async def score_chain(chain):
              context = [
                  {"name": "Problem", "data": problem, "description":
↪ "The original problem statement."},
                  {"name": "Chain",    "data": str(chain),
↪ "description": "Full chain-of-thought reasoning plus final
↪ answer."}
              ]
              instructions = (
                  "You are evaluating the chain-of-thought solution for
↪ the given problem. "
                  "On a scale from 1 (very uncertain) to 5 (very
↪ confident), rate your confidence "
                  "that the final answer is correct. Output ONLY the
↪ integer confidence (1-5)."
              )
              response_format = [{"name": "Confidence", "description":
↪ "Integer from 1 to 5"}]
              resp = await self.llm.chat_completion(context,
↪ instructions, response_format)
              # parse safely
              try:
                  conf = int(resp["Confidence"].strip())
              except Exception:
                  conf = 1
              return max(1, min(conf, 5))

          # Helper: verify logical consistency (Yes/No)
          async def verify_chain(chain):
              context = [
                  {"name": "Problem", "data": problem, "description":
↪ "The original problem statement."},
                  {"name": "Chain",    "data": str(chain),
↪ "description": "Full chain-of-thought reasoning plus final
↪ answer."}
              ]
              instructions = (
                  "Review the chain-of-thought reasoning for the given
↪ problem. "
                  "Is the reasoning free of logical errors or
↪ contradictions? "
                  "Output ONLY 'Yes' if it is fully logical, otherwise
↪ output 'No'."
              )
              response_format = [{"name": "Valid", "description": "Yes
↪ or No"}]
              resp = await self.llm.chat_completion(context,
↪ instructions, response_format)
              valid = resp.get("Valid",
↪ "").strip().lower().startswith("y")
              return valid

          # Weighted vote helper
          def find_best_weighted(chains_list, conf_list):
              weight_sums = {}
              total = sum(conf_list)
              for chain, cf in zip(chains_list, conf_list):
                  ans = chain.answer()
                  weight_sums[ans] = weight_sums.get(ans, 0) + cf
              best_ans, best_w = None, -1
              for ans, w in weight_sums.items():
                  if w > best_w:
                      best_ans, best_w = ans, w
```

```python
70              return best_ans, best_w, total
71
72          # 1) Generate initial 3 chains in parallel
73          initial = [generate_chain() for _ in range(3)]
74          chains = await asyncio.gather(*initial)
75
76          # 2) Score and verify each chain
77          score_tasks = [score_chain(ch) for ch in chains]
78          verify_tasks = [verify_chain(ch) for ch in chains]
79          confidences = await asyncio.gather(*score_tasks)
80          valids = await asyncio.gather(*verify_tasks)
81
82          max_chains = 7
83
84          # 3) Adaptive sampling with verification gating
85          while True:
86              # Determine which chains to consider: only verified if
    ↪ any, else all
87              if any(valids):
88                  considered_chains = [ch for ch, v in zip(chains,
    ↪ valids) if v]
89                  considered_confs   = [cf for cf, v in
    ↪ zip(confidences, valids) if v]
90              else:
91                  considered_chains = chains
92                  considered_confs   = confidences
93
94              best_ans, best_weight, total_weight =
    ↪ find_best_weighted(considered_chains, considered_confs)
95              # stop if weighted majority reached or chain cap
96              if best_weight > total_weight / 2 or len(chains) >=
    ↪ max_chains:
97                  break
98
99              # else generate one more chain, score & verify, then loop
100             new_chain = await generate_chain()
101             chains.append(new_chain)
102             new_conf = await score_chain(new_chain)
103             confidences.append(new_conf)
104             new_valid = await verify_chain(new_chain)
105             valids.append(new_valid)
106
107         # 4) Select final chain: consensus & highest confidence among
    ↪ considered
108         if any(valids):
109             final_pool = [ (ch, cf) for ch, cf, v in zip(chains,
    ↪ confidences, valids) if v and judge_equality(ch.answer(),
    ↪ best_ans) ]
110         else:
111             final_pool = [ (ch, cf) for ch, cf in zip(chains,
    ↪ confidences) if judge_equality(ch.answer(), best_ans) ]
112
113         selected_chain = None
114         top_conf = -1
115         for ch, cf in final_pool:
116             if cf > top_conf:
117                 selected_chain, top_conf = ch, cf
118
119         # Fallback if nothing selected
120         if selected_chain is None:
121             selected_chain = chains[-1]
122
123         return selected_chain
```

Listing 2: Code for VerifiedWeightedAdaptiveSelfConsistentChainOfThought, performance: 38.0

## G  REPRODUCIBILITY STATEMENT

Upon publication, we commit to releasing the open-source code for our framework, including all discovered Agentic Reasoning Modules, meta-policies, and the specific prompts used for the Reviewer Agent. Our experiments were conducted using a mix of closed and open-source models. The MAS designer utilized OpenAI's `o4-mini-high` The reasoning modules were executed on `GPT-4.1-nano`, `GPT-4o`, and the open-source `Llama-3.3-70B`. All evaluation benchmarks, including MATH500, AIME, and HMMT, are publicly available.

### G.1  ARM IMPLEMENTATION DETAILS

The 1000-sample subset of Open-R1-Mixture-of-Thoughts was created by taking the math and science splits of the original dataset, filtering to samples which the provided Deepseek-R1 reasoning trace had length between 8k to 10k tokens (to filter to samples of appropriate difficulty), and randomly sampling 1000 problems from the filtered problems.

We run both the step-generator module optimization and the meta-policy optimization for 20 iterations. Both optimizations are performed using GPT-4.1-nano as the MAS executor model.

Whenever sampling from the MAS executor model, we use a temperature of 1.0 with a top_p of 0.95.

### G.2  BASELINE IMPLEMENTATION DETAILS

As in the ARM implementation, whenever sampling from the MAS executor model, we use a temperature of 1.0 with a top_p of 0.95.

- CoT: We use a simple CoT prompt that instructs the model to reason step-by-step and follow the final answer format.
- CoT-SC: We use $n = 12$ parallel reasoning traces.
- Self-Refine: We limit to a maximum of 5 self refining iterations.
- LLM-Debate: We use 4 LLM agents debating for a maximum of 3 rounds.
- ADAS: We use the provided codebase, following the recommended run configuration. For a fair comparison to other baselines, we make a one line addition to the optimizer prompt to disallow arbitrary Python code execution within the discovered MASes, since other baselines do not utilize code execution. For the 1000-sample optimization, we use GPT-4.1-nano as the MAS executor model during optimization, following ARM's implementation.
- AFlow: We use the provided codebase, following the recommended run configuration. We allow the optimizer to utilize the Custom, AnswerGenerate, and ScEnsemble operators. For the 1000-sample optimization, we use GPT-4.1-nano as the MAS executor model during optimization, following ARM's implementation.

## H  EXAMPLE OF REVIEWER FEEDBACK AND MUTATION

Here we show a concrete example of a reviewer surgical mutation, starting from an intermediate ARM module CriticChainOfThoughtV2, the reviewer feedback based on reflection on two rollouts, and corrective fixes to yield CriticChainOfThoughtV3.

```
ARM Mutation Example (CriticChainOfThoughtV2->CriticChainOfThoughtV3)

--- CriticChainOfThoughtV2.py
+++ CriticChainOfThoughtV3.py
import asyncio

class CriticChainOfThoughtV2:
    def __init__(self, llm):
        self.llm = llm
```

```
     async def forward(self, problem, partial_progress):
         # 1. Generate four candidate next steps in parallel
         candidate_tasks = [
             self.llm.generate_step(problem, partial_progress)
             for _ in range(4)
         ]
         candidates = await asyncio.gather(*candidate_tasks)

         # 2. Critique candidates in two groups of two, in parallel
         ...
         # 3. Parse ratings and identify the two highest-rated
         ↪  candidates
         ...

         # 4. Final head-to-head comparison between the top two
         ↪  candidates
         ...
-        # 5. Return the selected candidate
-        if final_decision["winner"].strip() == "Candidate A":
-            return top1_candidate
-        else:
-            return top2_candidate
+        if final_decision["winner"].strip() == "Candidate A":
+            selected_candidate = top1_candidate
+            runnerup_candidate = top2_candidate
+        else:
+            selected_candidate = top2_candidate
+            runnerup_candidate = top1_candidate
+
+        # 5. Post-selection adversarial critique of the chosen
↪  step
+        context_flaw = [
+            {
+                "name": "Problem",
+                "data": problem,
+                "description": "The problem to solve."
+            },
+            {
+                "name": "Partial Progress",
+                "data": partial_progress,
+                "description": "The partial solution so far."
+            },
+            {
+                "name": "Selected Candidate Next Step",
+                "data": f"### Selected Candidate Next
↪  Step\n{selected_candidate}",
+                "description": "The final chosen candidate next
↪  reasoning step formatted with a markdown subheader."
+            }
+        ]
+        instructions_flaw = (
+            "You are given a problem, the current partial
↪  solution, and a selected next reasoning step.\n"
+            "Identify any major flaw or missing piece of reasoning
↪  in the selected step.\n"
+            "Provide:\n"
+            "- flaw: either the single word 'None' if there is no
↪  flaw, or a brief description of the flaw."
+        )
+        response_format_flaw = [
+            {
+                "name": "flaw",
```

```
+                    "description": "Either the single word 'None' if
↪    there is no flaw, or a brief description of a major flaw in the
↪    selected step."
+                }
+           ]
+           flaw_response = await self.llm.chat_completion(
+                context_flaw, instructions_flaw, response_format_flaw
+           )
+           flaw = flaw_response["flaw"].strip()
+
+           # 6. If a flaw is detected, fall back to the runner-up;
↪    otherwise keep the selected candidate
+           if flaw.lower() != "none":
+                return runnerup_candidate
+           return selected_candidate
```

## Reviewer Feedback on CriticChainOfThoughtV2

```
Performance Review of CriticChainOfThought    CriticChainOfThoughtV2

Overall Performance Change
- CriticChainOfThought achieved an average score of 36.1, while
    CriticChainOfThoughtV2 reached 37.1 (an absolute gain of +1.0).
- This gain indicates that the added head-to-head comparison step in
    V2 yields a measurable improvement in next-step selection accuracy
     without disrupting baseline reasoning quality.

Evidence from the Execution Traces

1. Trace 1 (Divisibility Problem)
        Both the previous block and V2 produced virtually identical
         Step 1   analyses, correctly identifying the need to
        enforce divisibility and positivity constraints.
         V 2 s  head-to-head mechanism did not reduce the clarity or
        correctness of the initial reasoning step, demonstrating that
        the extra selection stage does not degrade generation quality.

2. Trace 2 (Complex Numbers, Roots of Unity)
        Both blocks ultimately concluded that the set \(S\) must be the
         \(n\)th roots of unity for \(n=3,4,5\), and both enclosed the
        final answer in `<final_answer>`.
        V2 preserved the previous b l o c k s  correct reasoning while
        adding no spurious steps, confirming that the new comparison
        stage retains core functionality.

3. Trace 3 (Consecutive Hits Combinatorics)
         The previous b l o c k s  initial steps vacillated between
        interpreting  6 .8   as 6, 7, or 8 shots across multiple
        retries.
         In contrast, V 2 s   Step 1    consistently assumes 8 shots
        and clearly restates the pattern constraints before counting,
        suggesting that the head-to-head comparison helped V2 converge
        on a single, more accurate interpretation of the problem
        statement.

How the Evolution Drove the Improvement
- The original block generated diverse candidates and used two-way
    critiques to pick a top-rated step, but sometimes retained
    ambiguous or less precise interpretations.
- By introducing a third, final head-to-head comparison between the
    two frontrunners, V2 more reliably resolves borderline
     c a s e s such as exactly how many total shots to assume in the
    combinatorics p r o b l e m leading to slightly better overall
    accuracy.
- Crucially, this additional decision stage does not interfere with
    the correctness of initial reasoning steps or final answers, as
    seen in Traces 1 and 2, but does reduce interpretation noise in
    tricky examples like Trace 3.

Conclusion
CriticChainOfThoughtV2s addition of a head-to-head decision step
    raises next-step selection accuracy by resolving close calls
    between top candidates, yielding a +1.0 performance l i f t evident
     in more consistent problem interpretationswhile fully
    preserving the original b l o c k s  reasoning quality.
```

```
Reviewer Feedback on CriticChainOfThoughtV3

Performance Summary
The core change from CriticChainOfThoughtV2 to V3 was the addition of
    a post selection adversarial critique: after choosing between
    the top two candidate steps, V3 asks  is  there any major flaw?
        and, if the critic finds one, it falls back to the runner-up.
    In trace 1 (the quantum-mechanics problem) we see that
- V2 stopped at Step 3 without producing the required final answer (
    violating the  if  complete, immediately give <final_answer>
    rule).
- V3 both produced the correct reasoning step and appended the final
    answer `<final_answer>B</final_answer>`.

However, over the full benchmark of challenging reasoning problems,
    V3s  overall next-step selection accuracy dipped slightly (from
    37.1% to 36.7%). This suggests that while the added adversarial
    critique can catch some subtle mistakes (and in the one provided
    trace corrected  V2s  omission of the final answer), it also
    sometimes overrules a correct choice leading to more false
    negatives than V2.

Key Observations
1. Completeness: V3 correctly obeyed the    completefinal    answer
     rule in trace 1; V2 did not.
2. Average Accuracy: Despite the extra safeguard,  V3s  head-to-head
    + adversarialfallback pipeline results in a small net decrease
    in average next-step accuracy.
3. Critique Calibration: The adversarial critic occasionally
    identifies   flaws   where there are none, triggering
    unnecessary fallbacks.

Recommendations for V4
    Loosen the fallback criterion or better calibrate the critics
    prompt so it only flags genuine errors.
    Consider logging how often the post-selection critique actually
    flips the choice, and analyze those cases to refine the critique
    instructions.
    Explore integrating a lighterweight self-check earlier (e.g.
    during rating) rather than as a hard post-selection veto.
```

## I  SEED COT IMPLEMENTATION

The following is the Python implementation of the seed CoT step-generator that we use as the seed module and the corresponding prompt.

```python
class ChainOfThought:
    def __init__(self, llm):
        self.llm = llm

    async def forward(self, problem, partial_progress):
        generated_step = await self.llm.generate_step(problem,
    partial_progress)
        return generated_step
```

Listing 3: Code for CoT Seed

**CoT Prompt (Step-Generator)**

```
## SYSTEM TURN
You will be tasked to complete a partial solution to a problem.

Continue the partial solution in a step by step manner, denoting the
    first step with "### Step {n}" and so on.

Each step should significantly progress the partial solution, without
    repeating any work done in previous steps.

Once the solution is completed, your response should conclude by
    giving the final answer enclosed within <final_answer> </
    final_answer>.

Do not include units in the final answer. For multiple choice
    questions, the final answer should only be the letter of the
    correct answer choice.

If the partial solution is already complete, immediately provide the
    final answer instead of writing another step.

## USER TURN
Problem:

{problem}

Complete this partial solution to the problem:

{partial_progress}
```

