# OpenReview forum: "ARM: Discovering Agentic Reasoning Modules for Generalizable Multi-Agent Systems"
_ICLR.cc/2026/Conference — ICLR 2026 Conference Desk Rejected Submission_

### Official Review · Reviewer_T74Q · 2025-10-15

**Soundness:** 3
**Presentation:** 2
**Contribution:** 2
**Rating:** 2
**Confidence:** 3

**Summary:**

This paper identifies a critical issue in current research: complex, automatically-designed Multi-Agent Systems often fail to outperform simple CoT baselines. The authors argue that the primary bottleneck is not the high-level agent orchestration but the quality of the fundamental, step-by-step reasoning unit. They propose the ARM, a self-contained, code-based agentic system that serves as a powerful, "drop-in" replacement for a single CoT step. This ARM module is not manually engineered but is automatically discovered through a reflection-guided evolutionary search that starts with a basic CoT implementation and iteratively mutates it based on performance feedback. To manage computational complexity, the framework is cleverly decoupled: an optimal step-generator module and an optimal high-level meta-policy are discovered independently. The meta-policy search uses the computationally cheap baseline CoT module as a surrogate, with the resulting policy effectively transferring to the more powerful ARM.

**Strengths:**

1. The paper's core premise—that improving the foundational reasoning step is more fruitful than designing ever-more-complex agent superstructures—is a crucial and timely contribution to the field.
2. The empirical and theoretical justifications for this transfer add significant rigor.
3. Comprehensive results support the work.

**Weaknesses:**

1. While the method discovers a superior ARM, the paper does not provide sufficient insight into what makes the final module better. What specific agentic patterns (e.g., self-reflection loops, parallel hypothesis generation, weighted aggregation) are consistently evolved?
2. The paper does not quantify the computational cost of the discovery process (e.g., total tokens, GPU hours, or monetary cost).
3. The success of the reflection-guided search is contingent upon the high-level reasoning and coding capabilities of the "Reviewer Agent". The paper does not explore the sensitivity of the discovery process to the choice of this designer LLM.

**Questions:**

1. Could you provide the final, discovered Python code for the top-performing ARM module and its corresponding Meta-Policy? Seeing the concrete implementation would offer profound insight into the "agentic block" that replaces the CoT step.
2. The scaffolded objective uses a fixed window size of l=3 steps for evaluating candidate modules. How was this value chosen? Have you analyzed the sensitivity of the final discovered module's performance to this hyperparameter?
3. The meta-policy is discovered using mCoT as a surrogate, which may bias the search toward strategies that are optimal for a simple, recursive step generator. Does this approach risk missing out on more complex strategies (e.g., those involving dynamic branching or resource allocation) that might be uniquely effective when orchestrating the more powerful m∗ module?
4. Given ARM's success in revitalizing a fundamental reasoning primitive, have you considered its application beyond question-answering tasks? For instance, could it be adapted to enhance steps within other sequential processes like code generation, mathematical theorem proving, or long-term planning?

---

> ### Author Response · Authors · 2025-11-24
> **Rebuttal by Authors [1/3]**
>
> We want to thank the reviewer for their feedback. We appreciate the reviewer’s positive comments on our contribution exploring a critical gap in reasoning MAS systems. Here we address the weakness feedback and answer the questions.
>
> **Response to W1)**
>
> We have updated the paper to provide the concrete code for the discovered ARM module and the meta policy, along with example ARM rollouts as well as examples of Reviewer’s mutation strategy. We agree that including these artifacts in the appendix will provide good insight into what the discovered ARM module is doing. We also explain these below:
>
> **ARM:** The top-performing ARM module (named CriticChainOfThoughtV7) starts by speculating four candidate solutions, performs pair wise comparison and critique, identifies the top two winners, and performs a head-to-head comparison of them to pick a winner, followed by a final adversarial critique and a dynamic threshold-based fallback. Effectively, the module integrates deliberation, verification, self-consistency, and risk sensitive gating all within a single reasoning step.
>
> **Meta Policy:** The learned meta policy is quite sophisticated, adopting techniques such as adaptive self-consistency, verification gated sampling, and confidence weighted consensus over reasoning chains. Instead of committing to a fixed number of rollouts, it dynamically generates new chains only when the current ones fail to reach a weighted majority, using both confidence scores (1–5) and a separate logical-validity check to decide which chains to trust. When a majority is reached, the policy selects the best chain supporting that answer, preferring those that are both valid and highly confident. This combination of uncertainty estimation, logical verification, dynamic stopping, and consensus selection is a non-trivial structure for manual engineering.
>
> ---
>
> **Response to W2)**
>
> Thank you for this important question. We have added the computation cost (from OpenAI API) analyses below.
>
> There are two costs to discuss: i) search and ii) inference. We show that a) ARM is substantially more amortization-efficient during search compared to baselines ADAS and AFlow which need domain and model specific tuning, and b) among multi-agentic systems which are necessarily costly, only ARM delivers positive marginal returns with increasing compute.
>
> **W1.1. Search Cost**
>
> **A) Training cost per iteration on 1k training samples (GPT-4.1-mini as executor and o4-mini as Reviewer):**
>
> | Model            | Cost (USD)                         |
> |------------------|------------------------------------|
> | ARM (Ours)       | 4.53                               |
> | Meta (Ours)      | 5.40                               |
> | **Total Cost (Ours)** | **9.93** (ARM + Meta: 4.53 + 5.40) |
> | ADAS             | 2.62                               |
> | AFlow            | 2.52                               |
>
> While the ARM policy search cost in a single iteration is 1.72x that of ADAS and 1.79x that of AFlow, this cost need to be amortized by domain and model. Both ADAS and AFlow are domain and model specific, and for the models and benchmarks we consider, the search cost is multiplied by a factor of 12 (4 domains * 3 models). In contrast, ARM generalizes across domains and models (Lines 351-353). Thus, the amortized cost for ARM is only 0.14x of ADAS and 0.15x that of AFlow. For the ARM+Meta policy approach, the amortized cost is only 0.31x of ADAS and 0.32x of AFlow.
>
> **W1.2. Inference Cost**
>
> **Inference Cost (USD) on 1K samples of OpenReasoner dataset (GPT-4.1-nano):**
>
> | Method           | AIME | HMMT | GPQA | LiveBench | Avg  |
> |------------------|------|------|------|-----------|------|
> | ARM (Ours)       | 3.69 | 3.02 | 0.81 | 1.12      | 2.16 |
> | ARM + Meta (Ours)| 17.60| 13.77| 3.22 | 4.36      | 9.73 |
> | ADAS             | 0.87 | 0.88 | 0.19 | 0.28      | 0.55 |
> | AFlow            | 0.78 | 0.66 | 0.16 | 0.19      | 0.45 |
> | CoT              | 0.19 | 0.18 | 0.06 | 0.11      | 0.13 |
>
> ARM is intentionally a “compute-for-performance” training approach: rather than running a workflow once for the entire reasoning task, which is prone to error propagation, we run the ARM reasoning module at each granular step, trading compute for performance. This opens a new frontier in reasoning research with automatically discovered Multi-Agentic systems.
>
> **Crucially**, our architecture does allow for cost-optimized ARM variants. Since ARM is iteratively improved using a GEPA (Agrawal et.al) style reflection guided search, a cost term to the reviewer agent directly yields cost-performance pareto variants. We leave this for future work and will clarify feasibility in the final version of the paper.

---

> > ### Author Response · Authors · 2025-11-24
> > **Rebuttal by Authors [2/3]**
> >
> > **W1.3. Generalizability of ADAS and AFlow across domains/models**
> >
> > For fair comparison, we also investigated the cross-domain applicability of AFlow and ADAS by training them on the same 1K sample split from OpenReasoner dataset as ARM, and we observe poor cross-domain performance transfer. Even allowing for *model specific* training on the 1K train set, did not yield performance gains on other domains using the same model. This strongly indicates that these methods are brittle and cannot leverage large-scale datasets beyond the limited per-benchmark validation sets.
> >
> > | Method (1K train) | Model          | AIME | HMMT | GPQA | LiveBench |
> > |-------------------|----------------|------|------|------|-----------|
> > | ADAS              | GPT-4.1-mini   | 0.0  | 6.8  | 46.8 | 29.4      |
> > | ADAS              | GPT-4o         | 0.0  | 0.5  | 46.8 | 41.9      |
> > | ADAS              | Llama-3.3-70B  | 3.1  | 0.5  | 42.4 | 46.2      |
> > | AFlow             | GPT-4.1-mini   | 16.7 | 10.4 | 51.3 | 30.6      |
> > | AFlow             | GPT-4o         | 9.4  | 0.0  | 50.6 | 45.0      |
> > | AFlow             | Llama-3.3-70B  | 7.2  | 3.1  | 46.8 | 15.6      |
> >
> > ---
> >
> > **Response to W3)**
> >
> > The reviewer module is indeed crucial for guiding the ARM and Meta Policy generation and follows a GEPA (Agrawal et al.) style optimization, but on the agentic system (represented in code) itself rather than just the prompts. For this reason, we used a strong reasoning (thinking) model in this stage that has achieved best performance on code understanding, generation, and reasoning benchmarks.
> > We had also tried GPT-4.1-mini and GPT-4.1-nano, but observed noticeable issues in their verbal feedback.
> >
> > ---
> >
> > **Response to Questions:**
> >
> > **Q1:** We have updated the paper to provide the concrete code for the discovered ARM module in Appendix-D (ARM module titled **CRITICCHAINOFTHOUGHTV7**) and the meta policy in Appendix-E, along with ARM rollout examples on the following pages.
> >
> > For further details, we kindly refer the reviewer back to the response for Weakness 1.
> >
> > **Q2:** Thank you for this clarification question. We pick the hyperparameter *l* based on theoretical insight. Our goal is to train in a scaffolded environment with the lowest number of ARM rollouts, while still allowing the three possible configurations needed to preserve state transition dynamics between CoT and ARM:
> >
> > - CoT output → ARM input
> > - ARM output → ARM input
> > - ARM output → CoT input
> >
> > We chose *l = 3* as it is the minimal window size that supports these three configurations. We have clarified this in the updated manuscript. In the final version, we will add an ablation study varying *l* (within context-length limits) and analyzing its impact on the performance of the discovered ARM.
> >
> > **Q3:** We thank the reviewer for this thoughtful question. Our design explicitly prevents the meta-policy from being restricted to “what works for mCoT.” The scaffolded training deliberately ensures that the discovered ARM module \(m^*\) is a strict drop-in refinement of the CoT operator rather than a qualitatively new operator. This is motivated by the observation that CoT has high performance and is the core reasoning engine in all MAS strategies, including hand-engineered methods (Debate, Self-Consistency, etc.) as well as automatic methods such as ADAS/AFlow. Thus, meta-policies learned on CoT-scaffolded traces can transfer well to a strictly better granular reasoning module such as ARM. Additionally, as noted in Section 3.2, the scaffolded training offers advantages such as direct credit assignment during the search phase and acts as an inductive bias to search for granular reasoning modules.
> >
> > To analyze this further, we have expanded the theory section (Appendix-B), investigating conditions under which such off-policy transfer is valid. We show that the scaffolded objective minimizes per-step error rate (Proposition 1), improves induced states (i.e., accumulated partial progress; Proposition 2), and induces a beneficial state distribution (Proposition 3).
> >
> > Importantly, as seen in Appendix-C of the updated manuscript, the discovered meta policy already includes dynamic branching, parallel rollouts, verification gating, adaptive sample allocation, and confidence-weighted aggregation—directly addressing the reviewer’s concerns. These orchestration patterns amplify the benefits when used with a superior ARM module compared to CoT and do not miss out due to scaffolded training.

---

> ### Author Response · Authors · 2025-11-24
> **Rebuttal by Authors [3/3]**
>
> **Q4:** Thank you for recognizing the applicability of ARM as a general reasoning primitive that can be used in many sequential tasks like coding, theorem proving, or long-term planning. Nothing in the design limits it to question answering. For this paper, however, we purposely chose domains where evaluation is clean, automatic, and does not require extra scaffolds. Math and QA benchmarks let us directly measure correctness and per-step error rates, which are essential for our theoretical analysis and for comparing modules fairly. In contrast, coding requires compilers and test harnesses, theorem proving often needs proof assistants or manual checking, and long-term planning depends on simulators or complex environments. These extra systems would introduce noise and confounding factors that make it harder to isolate the reasoning ability of the step-module itself. The four challenging benchmarks we use already provide strong, diverse, and high-signal evidence that ARM discovers a strictly better reasoning step and that this improvement transfers across tasks and models.

---

> ### Author Response · Authors · 2025-11-27
> **Request for reviewer feedback on the Rebuttal**
>
> Dear Reviewer,
>
> We thank you again for the detailed feedback. We have posted our rebuttal along with substantial appendix updates addressing each  of the concerns raised in the review. These include the discovered ARM and Meta Policy modules and their analysis, cost accounting, convergence analysis, insights into the reviewer module, along with strengthened theoretical analysis to justify off-policy transfer of the meta policy. We have also responded to all the questions including the choice of hyperparameter $l$ and applicability of ARM to other domains.
>
> We believe these updates fully resolve the points discussed in the review. We are eagerly looking forward to hearing from you.
>
> Sincerely,
>
> Authors

---

### Official Review · Reviewer_5da4 · 2025-10-30

**Soundness:** 3
**Presentation:** 2
**Contribution:** 3
**Rating:** 4
**Confidence:** 3

**Summary:**

This paper introduces an agentic reasoning system comprising the low-level Agentic Reasoning Module (ARM) and the top-level Meta Policy. The ARM is discovered via evolutionary search, aiming to optimize the quality of single-step CoT reasoning. The Meta Policy learns how to orchestrate the execution flow of the ARM to solve complex, multi-step global reasoning tasks.

**Strengths:**

The adoption of a reflection-guided evolutionary search for automatically discovering both ARM and the Meta Policy is a methodological contribution, eliminating the need for manual engineering of complex, heterogeneous MAS. Experimental results demonstrate that both ARM and the combined ARM+Meta Policy achieve better performance across diverse task domains.

**Weaknesses:**

1. The proposed method might lead to complex workflows, which inherently suggests high computational overhead. While the paper fails to provide quantified data in the main text on the actual inference cost (i.e., total token consumption, API call count, or average latency) compared to efficient baselines like CoT or CoT-SC. The performance gains must be explicitly benchmarked against the running cost to justify their practicality in real-world applications.

2. The paper lacks a thorough investigation into the stability of the optimization process: a) seed sensitivity: How sensitive are the final discovered modules to the initial CoT implementation and the specific sampling of the training trajectories? b) convergence analysis: A search trajectory to demonstrate the robustness and efficiency of the search algorithm is needed. Without this, the reliability of the discovered architecture remains questionable.

3. Both the ARM and the Meta Policy are discovered as complex code-based workflows. Compared to manually designed MAS (like LLM-Debate or CoT-SC), which have clear, human-understandable agent roles, the interpretability of the discovered architecture is a major concern. If the complex workflow logic is opaque, it severely limits the system's applicability in domains requiring verifiable and trustworthy reasoning.

**Questions:**

1. What was the total computational budget consumed for discovering both ARM and Meta Policy?

2. Could the author provide some concrete examples of the Meta Policy's dynamic control flow? Under what conditions does the Meta Policy choose to initiate a branching parallel search versus conditional loops for verification?

3. How does meta policy determine the termination condition? Does the discovered meta policy incorporate an explicit backtracking mechanism, and if so, how is the trigger and execution of the backtrack handled?

4. The authors are encouraged to discuss more about the bad case of the proposed method.

---

> ### Author Response · Authors · 2025-11-24
> **Rebuttal by Authors [1/3]**
>
> We sincerely appreciate the reviewer’s encouraging feedback and insightful questions. Our detailed responses to each concern are provided below.
>
> **Response to Weakness 1)**
>
> We agree that cost analysis is important for practicality. We have therefore added cost analysis for both i) search and ii) inference and show that a) ARM is substantially more amortization-efficient during search compared to baselines ADAS and AFlow which need domain and model specific tuning, and b) among multi-agentic systems which are necessarily costly, only ARM delivers positive marginal returns with increasing compute. We discuss search and inference costs below.
>
> **W1.1. Search Cost**
>
> **A) Training cost per iteration on 1k training samples (openr1/mixed-thoughts dataset with GPT-4.1-mini):**
>
> | Model            | Cost (USD)                         |
> |------------------|------------------------------------|
> | ARM (Ours)       | 4.53                               |
> | Meta (Ours)      | 5.40                               |
> | **Total Cost (Ours)** | **9.93** (ARM + Meta: 4.53 + 5.40) |
> | ADAS             | 2.62                               |
> | AFlow            | 2.52                               |
>
> While the ARM policy search cost in a single iteration is 1.72× that of ADAS and 1.79× that of AFlow, this cost needs to be amortized by domain and model. Both ADAS and AFlow are domain and model specific, and for the models and benchmarks we consider, the search cost is multiplied by a factor of 12 (4 domains × 3 models). In contrast, ARM generalizes across domains and models (Lines 351–353 of PDF). Thus, the amortized cost for ARM is only **0.14×** of ADAS and **0.15×** of AFlow. For the ARM+Meta policy approach, the amortized cost is **0.31×** of ADAS and **0.32×** of AFlow.
>
> **W1.2. Inference Cost**
>
> **Inference Cost on GPT-4.1-nano (USD):**
>
> | Method           | AIME | HMMT | GPQA | LiveBench | Avg  |
> |------------------|------|------|------|-----------|------|
> | ARM (Ours)       | 3.69 | 3.02 | 0.81 | 1.12      | 2.16 |
> | ARM + Meta       | 17.60| 13.77| 3.22 | 4.36      | 9.73 |
> | ADAS             | 0.87 | 0.88 | 0.19 | 0.28      | 0.55 |
> | AFlow            | 0.78 | 0.66 | 0.16 | 0.19      | 0.45 |
> | CoT              | 0.19 | 0.18 | 0.06 | 0.11      | 0.13 |
>
> ARM is intentionally a “compute-for-performance” training approach: rather than running a workflow once for the entire reasoning task, which is prone to error propagation, we run the ARM reasoning module at each granular step, trading compute for performance. This opens a new frontier in reasoning research with automatically discovered Multi-Agentic systems.
>
> **Crucially**, our architecture does allow for cost-optimized ARM variants. Since ARM is iteratively improved using a GEPA (Agrawal et al.) style reflection-guided search, a cost term to the reviewer agent can yields cost-performance Pareto variants. We leave this for future work.
>
> **W1.3. Generalizability of ADAS and AFlow across domains/models**
>
> For fair comparison, we also investigated the cross-domain applicability of AFlow and ADAS by training them on the same 1K sample split from the OpenReasoner dataset as ARM, and we observe poor cross-domain performance transfer. Even allowing for *model specific* training on the 1K train set did not yield performance gains on other domains using the same model. This strongly indicates that these methods are brittle and cannot leverage large-scale datasets beyond the limited per-benchmark validation sets.
>
> | Method (1K train) | Model          | AIME | HMMT | GPQA | LiveBench |
> |-------------------|----------------|------|------|------|-----------|
> | ADAS              | GPT-4.1-mini   | 0.0  | 6.8  | 46.8 | 29.4      |
> | ADAS              | GPT-4o         | 0.0  | 0.5  | 46.8 | 41.9      |
> | ADAS              | Llama-3.3-70B  | 3.1  | 0.5  | 42.4 | 46.2      |
> | AFlow             | GPT-4.1-mini   | 16.7 | 10.4 | 51.3 | 30.6      |
> | AFlow             | GPT-4o         | 9.4  | 0.0  | 50.6 | 45.0      |
> | AFlow             | Llama-3.3-70B  | 7.2  | 3.1  | 46.8 | 15.6      |
>
> ---
>
> **Response to Weakness 2)**
>
> Due to computational constraints, we ran a single full search on a randomly sampled 1k subset of the OpenReasoner dataset. To probe training stability, we evaluated multiple intermediate ARM variants along the trajectory and observed smooth performance improvements on the GPQA dataset, without any sudden performance collapses. We have added this figure to the appendix (See Figure-4). The top-10 discovered modules all lie within 5% of each other, suggesting low variance w.r.t. the particular samples shown to the reviewer.
>
> There are two things to note:
> a) the search space is not continuous, but discrete since the next MAS iteration is proposed by an LLM-based reviewer agent.
> b) the temperature-sampling search procedure is tree-based (exploration/exploitation) and naturally produces bumpy trajectories as it occasionally explores lower-performing modules.

---

> > ### Author Response · Authors · 2025-11-24
> > **Rebuttal by Authors [2/3]**
> >
> > **Response to Weakness 3)**
> >
> > We appreciate the reviewer’s concern regarding interpretability. We respectfully clarify that all code-based automatic MAS construction methods—including ADAS, AFlow, MaAS, and our proposed ARM—produce complex code-based workflows whose internal prompts and structures are not human-engineered. In this shared paradigm, ARM is in fact the most interpretable in both search and inference. Additionally, ARM provides interpretability advantages over human-engineered methods because of granular CoT-like outputs, offering a direct window into incorrect reasoning steps when they occur.
> >
> > **Search:** Existing automatic MAS methods such as ADAS and AFlow evolve large agentic workflows using validation-only feedback: each mutation is proposed solely to improve a scalar accuracy score. In ADAS, for example, the update prompt instructs the LLM to modify the agentic system to “maximize fitness” and “recall the agentic literature” [1], without providing structured analysis of why the previous workflow failed. This makes the mutation process difficult to audit and the resulting architectures opaque.
> >
> > In contrast, ARM’s discovery loop is explicitly reflection-driven. The Reviewer module inspects execution traces, identifies concrete failure modes, pinpoints which lines or behaviors caused those failures, and proposes a targeted code-level fix. This yields a transparent reasoning chain for each mutation:
> > i) which rollout behaviors were problematic,
> > ii) how the Reviewer diagnosed them,
> > iii) what precise changes were made and why.
> >
> > We include an example Reviewer feedback and mutation in Appendix-G showing (a) the sampled ARM module, (b) the Reviewer’s identified weaknesses, and (c) the resulting targeted modification.
> >
> > **Inference:** During inference, ARM provides another dimension of interpretability because its outputs are CoT-like but include an additional reliability layer (scoring, comparison, flaw detection). Thus, ARM’s transcripts are far more interpretable than the multi-turn discussions in LLM-Debate or the multi-agent message chains in ADAS/AFlow. This problem is more critical in the newly introduced baseline (based on reviewer-dWxr’s suggestion) where the MAS is produced by an autoregressive neural network controller, limiting interpretability regarding why an operator was selected.
> >
> > We have added a side-by-side comparison of ARM and CoT reasoning outputs in Appendix-C. Additionally, ARM’s internal logs (not part of the final output) preserve reasoning about why a particular branch was chosen and its confidence metrics; we will make this clearer in the final draft. The meta-policy, though complex, is also interpretable: Appendix-E shows that it launches three chains, scores them (1–5), verifies them via a binary gate, and repeatedly performs weighted majority voting until >50% consensus or the chain-budget threshold is reached. Its termination/branching conditions are explicit arithmetic comparisons in the code, not learned weights as in MaAS (Zhang et al.).
> >
> > ---
> >
> > **Response to Questions:**
> >
> > **Q1:** Addressed in response to W1. We run the ARM and Meta Policy search for 20 iterations, representing a total cost of 20 × 4.53 = **$90.60** for ARM and 20 × 5.40 = **$108** for Meta Policy using OpenAI o4-mini rates. Total: **$198**. Peak performance is observed mid-search (iterations 8–10), as shown in Appendix-H.
> >
> > **Q2:** We have now provided the full code for the discovered ARM and Meta Policy modules in Appendix-D and Appendix-E, respectively. These artifacts give full insight into the modules. We answer the sub-questions here:
> >
> > **When does the meta-policy do parallel branching?**
> > At the beginning, it always starts with three parallel chains:
> >
> > initial = [generate_chain() for _ in range(3)]
> > chains = await asyncio.gather(*initial)
> >
> >
> > Later in the loop, it may branch further by adding chains one at a time until the confidence threshold is reached:
> >
> > if best_weight > total_weight / 2 or len(chains) >= max_chains:
> > break
> >
> > new_chain = await generate_chain()
> > chains.append(new_chain)
> >
> >
> > **When does the policy use conditional loops?**
> > On each iteration, the policy recomputes “eligible” chains using a verification gate:
> >
> > if any(valids):
> > considered_chains = [ch for ch, v in zip(chains, valids) if v]
> > considered_confs = [cf for cf, v in zip(confidences, valids) if v]
> > else:
> > considered_chains = chains
> > considered_confs = confidences
> >
> >
> > If any chain passes verification, the loop uses only verified chains for the weighted vote; otherwise it falls back to all chains.
> >
> > **Summary:**
> > The meta-policy begins with parallel branching (three chains), always runs verification inside the loop, expands branching only when consensus is weak and budget allows, and accepts verification only when at least one chain is logically valid.

---

> > > ### Author Response · Authors · 2025-11-24
> > > **Rebuttal by Authors [3/3]**
> > >
> > > **Q3:** The search converged on performing reasoning-step backtracking within the ARM module rather than in the meta-policy. However, the meta-policy performs “chain-level” backtracking by:
> > > i) exploring several (dynamically chosen) paths,
> > > ii) down-weighting/discarding chains via self-consistency verification,
> > > iii) adaptively sampling more chains until consensus is reached.
> > >
> > > The ARM module itself performs step-level backtracking in the adversarial flaw-check stage, discarding the winner and selecting the runner-up when flaw estimates are high.
> > >
> > >
> > >
> > > **Q4:** We agree that discussing failure modes is important. ARM and ARM+Meta are most beneficial on difficult, multi-step reasoning tasks where CoT has a non-trivial error rate. On very easy problems, the additional sampling/verification overhead can be unnecessary, and we sometimes observe small regressions relative to CoT due to noise in confidence/validity judges. In latency-critical applications, multi-chain inference may also be too expensive unless run in cost-constrained modes (e.g., smaller max_chains, fewer ARM steps) or trained with an additional cost objective (see W1.2). We will add a subsection summarizing these bad cases and clarify that ARM provides the most value in high-accuracy, high-stakes settings.
> > >
> > > ---
> > >
> > > [1] https://github.com/ShengranHu/ADAS/blob/2702bee8fefda42255efc5be9f60e3bd3db96ae4/_gpqa/gpqa_prompt.py#L495

---

> ### Author Response · Authors · 2025-11-27
> **Request for reviewer feedback on rebuttal**
>
> Dear Reviewer,
>
> We thank you again for the detailed feedback. We have posted our rebuttal along with substantial appendix updates addressing each  of the concerns raised in the review. These include cost analysis, convergence analysis, comments on interpretability, rollout examples, and the discovered ARM and Meta Policy modules etc.
>
> We believe these updates fully resolve the points discussed in the review. We are eagerly looking forward to hearing from you.
>
> Sincerely,
>
> Authors

---

### Official Review · Reviewer_dWxr · 2025-10-31

**Soundness:** 2
**Presentation:** 2
**Contribution:** 2
**Rating:** 2
**Confidence:** 5

**Summary:**

This paper introduces the Agentic Reasoning Module (ARM), a novel approach for designing Multi-Agent Systems (MAS). The authors observe that complex, automatically designed MAS often underperform simple baselines like Chain-of-Thought (CoT). Their core idea is to improve the fundamental reasoning unit itself, rather than designing elaborate agent orchestration. ARM replaces each step in a CoT sequence with a specialized, agentic module discovered through an evolutionary, reflection-guided tree search over code space. This process separately optimizes a step-generator module ($m^*$) and a meta-policy ($\pi^*$). The resulting ARM-based systems are shown to be highly generalizable, achieving state-of-the-art performance across multiple reasoning benchmarks and foundation models without task-specific re-optimization.

**Strengths:**

1. The motivation of this paper is novel and provides new insights.

2. The paper is well-organized and uses language that is easy to understand.

3. The method has strong model generalization.

**Weaknesses:**

1. The experimental results are questionable: The paper's results (Table 1) show that CoT and CoT-SC often outperform ADAS and AFlow. However, ADAS is supposedly designed to optimize based on common reasoning modules like CoT and CoT-SC. Even if the optimization fails to find a better solution, its performance should theoretically be at least equal to that of CoT or CoT-SC, not worse. This makes me skeptical about the experimental setup and the validity of the results.

2. Lack of cost analysis: The experiments only analyze the performance of the final optimized agent.  There is no analysis of the computational cost incurred during the optimization process, nor is there an analysis of the execution cost (e.g., token usage) of the optimized agent itself.

3. Lack of comparison with stronger baselines: The paper did not compare against other relevant or potentially stronger baselines, such as MaAS[1].

Reference

[1]Zhang G, Niu L, Fang J, et al. Multi-agent Architecture Search via Agentic Supernet[C]//Forty-second International Conference on Machine Learning.

**Questions:**

1. Can the authors provide the code repository, prompts, or other additional experimental details to improve reproducibility?

2. Can the authors include a comparison against baselines such as MaAS[1]?

3. The paper mentions $l=3$ and only states that 'it works well', but it lacks a sensitivity analysis for this hyperparameter.

Reference

[1]Zhang G, Niu L, Fang J, et al. Multi-agent Architecture Search via Agentic Supernet[C]//Forty-second International Conference on Machine Learning.

---

> ### Author Response · Authors · 2025-11-23
> **Rebuttal by Authors [1/2]**
>
> We thank the reviewer for their thoughtful questions. Below we provide clarifications addressing each of the concerns.
>
> **Response to W1)** **Response to W1)** We respectfully disagree with this assessment and appreciate the chance to clarify. The assumption that ADAS and AFlow should at minimum match CoT or CoT-SC, since these seed operators are available to the search, does not necessarily hold in complex multi-step reasoning settings.
>
> - ADAS and AFlow perform evolutionary search guided solely by *validation set* performance, which can lead to overfitting on the validation set and underperformance on test set, especially on complex tasks where small validation set may not reflect the full complex reasoning distribution. Unlike datasets such as GSM8K, and MultiArith which are quite saturated (CoT performance 87% and 97% respectively), we evaluate on AIME, HMMT, GPQA and LiveBench which require deep multi-step reasoning and are prone to brittle error propagation in elaborate MAS structures. Thus, the search is *not guaranteed* to preserve or improve CoT’s reasoning performance on the test set. This has also been shown MAS-Zero (Ke et.al SEA@NeurIps 2025) and MaAS (Zhang et.al ICML-2025) where ADAS has lower performance than baseline CoT and other CoT derived operators (Self Consistancy, Debate etc). Our results show additionally that this trend also holds for AFlow as well, albeit to a lesser degree. In fact, as noted in lines 54-56 and empirically shown in Table-1, this forms the core motivation for our work where we aim to replace the CoT step with a multi-agentic system. This opens a new frontier in inference time scaling with agentic systems, by targeting granular reasoning steps.
>
> - We would like to clarify that we evaluated both ADAS and AFlow fairly, using their original code repositories and run configurations. We report the performance of the best performing (on validation set) discovered workflow, consistent with the methodology in original papers as well as MaAS.
>
> ---
>
> **Response to W2)** Thank you for raising this important point. There are two costs to discuss, i.e., search cost and inference cost.
>
> **Search Cost:**
>
> **A) Training cost per iteration on 1k training samples (GPT-4.1-mini):**
>
> | Model            | Cost (USD)                   |
> |------------------|------------------------------|
> | ARM (Ours)       | 4.53                         |
> | Meta (Ours)      | 5.40                         |
> | **Total Cost (Ours)** | **9.93** (ARM + Meta: 4.53 + 5.40) |
> | ADAS             | 2.62                         |
> | AFlow            | 2.52                         |
>
> While a single iteration of ARM + Meta policy is ~3.8x that of ADAS and ~3.9x that of AFlow, this cost must be interpreted correctly.
>
> - Domain and Model specific techniques such as ADAS, AFlow (and MaAS) must be run individually for each domain, and also for each domain-model combination, resulting in multiple runs for a diverse set of domains. Whereas ARM is both model and domain agnostic. Thus, with 4 domains and 3 models, cost of ADAS would become (4 × 3 × 2.62 =) **$31.44** and (4 × 3 × 2.52 =) **$30.24** for AFlow. On the other hand, amortized search cost for ARM remains the same and becomes ~0.31× of ADAS and ~0.32× of AFlow. This reduces further if more domains or models are considered.
>
> - For fair comparison, we also investigated the cross-domain applicability of AFlow and ADAS by training them on the same 1K sample split from the OpenReasoner dataset as ARM, and we observe poor cross-domain performance transfer. Even allowing for *model specific* training on the 1K train set did not yield performance gains on other domains using the same model. This strongly indicates that these methods cannot leverage large-scale datasets beyond the limited per-benchmark validation sets.
>
> | Method (1K train) | Model          | AIME | HMMT | GPQA | LiveBench |
> |-------------------|----------------|------|------|------|-----------|
> | ADAS              | GPT-4.1-mini   | 0.0  | 6.8  | 46.8 | 29.4      |
> | ADAS              | GPT-4o         | 0.0  | 0.5  | 46.8 | 41.9      |
> | ADAS              | Llama-3.3-70B  | 3.1  | 0.5  | 42.4 | 46.2      |
> | AFlow             | GPT-4.1-mini   | 16.7 | 10.4 | 51.3 | 30.6      |
> | AFlow             | GPT-4o         | 9.4  | 0.0  | 50.6 | 45.0      |
> | AFlow             | Llama-3.3-70B  | 7.2  | 3.1  | 46.8 | 15.6      |
>
> We again emphasize that each row is *model specific* but domain-agnostic MAS training, whereas ARM is both domain and model agnostic.
>
> **B) Inference Cost (USD) on 1K samples of OpenReasoner dataset (GPT-4.1-nano):**
>
> | Method           | AIME | HMMT | GPQA | LiveBench |
> |------------------|------|------|------|-----------|
> | ARM (Ours)       | 3.69 | 3.02 | 0.81 | 1.12      |
> | ARM + Meta       | 17.60| 13.77| 3.22 | 4.36      |
> | ADAS             | 0.87 | 0.88 | 0.19 | 0.28      |
> | AFlow            | 0.78 | 0.66 | 0.16 | 0.19      |

---

> > ### Author Response · Authors · 2025-11-23
> > **Rebuttal by Authors [2/2]**
> >
> > ARM is intentionally a “compute-for-performance” training approach. Rather than running a workflow once for the entire reasoning task, which is prone to error propagation, we run the ARM reasoning module at each granular step, trading compute for performance. This opens a new frontier in reasoning research with automatically discovered Multi-Agentic systems.
> >
> > **Crucially**, our architecture does allow for cost-optimized ARM variants. Since ARM is iteratively improved using a GEPA (Agrawal et al.) style reflection-guided search, adding a cost term to the reviewer agent directly yields cost-performance Pareto variants. We leave this for future work and will clarify feasibility in the final version of the paper.
> >
> > We also want to emphasize a key observation that ADAS, AFlow, and other MAS are substantially more expensive than simple CoT, yet lead to only marginal or no performance gain over CoT. Within the paradigm of (costlier) agentic systems, although ARM is more expensive, it consistently leads to much higher improvements. The ratio of performance gains over additional cost factor is better than ADAS and AFlow.
> >
> > **Cost vs Performance Comparison:**
> >
> > | Approach         | Cost multiple | Performance gain | Gain / cost |
> > |------------------|--------------:|-----------------:|-------------:|
> > | CoT              | 1.0 (0.13 $)  | -                | -            |
> > | ADAS             | 4.2           | -2.9             | -0.69        |
> > | AFlow            | 3.5           | -1.7             | -0.48        |
> > | ARM (Ours)       | 16.6          | +6.1             | +0.36        |
> > | ARM + Meta (Ours)| 74.8          | +11.2            | +0.14        |
> >
> > We have added this in the Analyses section of the Appendix (in the updated version of the paper), under the title SEARCH AND INFERENCE COST ANALYSIS.
> >
> > ---
> >
> > **Response to W3)** Thank you for this suggestion. We have now included MaAS results for GPT-4.1-Nano (best performing model) below as well as in the updated manuscript. Furthermore, as noted in W1, the MaAS paper also shows that ADAS achieves lower performance across the board compared to even the simplest CoT operator, which forms the core motivation for our work.
> >
> > **Benchmark Results (GPT-4.1-Nano):**
> >
> > | Benchmark | Accuracy |
> > |-----------|----------|
> > | AIME      | 19.4%    |
> > | HMMT      | 8.3%     |
> > | GPQA      | 57.5%    |
> > | LiveBench | 41.8%    |
> >
> > **Important conceptual clarification:** MaAS is fully orthogonal to ARM. MaAS optimizes multi-agent interaction topologies, whereas ARM optimizes the reasoning operator itself, representing a new lower-level frontier of granular reasoning operators.
> >
> > ---
> >
> > **Response to Questions:**
> >
> > **Q1:** We have updated the paper to include the discovered ARM module, the meta policy module, all prompts, examples of ARM run rollouts, and implementation details of our proposed method and baselines. These are all added in the Appendix of the updated paper. We fully commit to releasing the full code and all artifacts upon acceptance.
> >
> > **Q2:** Addressed in response to W3.
> >
> > **Q3:** We pick the hyperparameter *l* based on theoretical insight. Our goal is to train in a scaffolded environment with the lowest number of ARM rollouts, while still allowing the three possible state transition configurations between CoT and ARM:
> >
> > - CoT output → ARM input
> > - ARM output → ARM input
> > - ARM output → CoT input
> >
> > We chose *l = 3* as the minimal window size that supports all three configurations. In the final version, we will add a minor ablation over *l* within context-length limits, as space permits.
> >
> > ---
> >
> > [1] MAS-Zero: Designing Multi-Agent Systems with Zero Supervision
> > Zixuan Ke, Austin Xu, Yifei Ming, Xuan-Phi Nguyen, Caiming Xiong, Shafiq Joty
> > SEA@NeurIPS, 2025 (Oral)
> >
> > [2] Multi-agent Architecture Search via Agentic Supernet
> > Zhang, Guibin; Niu, Luyang; Fang, Junfeng; Wang, Kun; Bai, Lei; Wang, Xiang
> > ICML, 2025 (Oral)
> >
> > [3] GEPA: Reflective Prompt Evolution Can Outperform Reinforcement Learning
> > Lakshya Agrawal et al. (arXiv)

---

> > > ### Comment · Reviewer_dWxr · 2025-11-26
> > >
> > > I thank the authors for addressing some of my concerns and appreciate the additional effort put into the rebuttal. After carefully re-reading the paper and related works, I now acknowledge the validity of the paper's motivation.
> > >
> > > However, I have a few remaining observations and requests:
> > >
> > > 1.  **Regarding W3:** Since the authors claim that ARM is orthogonal to MaAS, could you provide experimental results showing the performance of ARM combined with MaAS?
> > > 2.  **Regarding the Method and Cost:** I appreciate the inclusion of the repository, prompts, and experimental details. However, upon examining the discovered `CriticChainOfThoughtV7` module, it appears to function as a workflow rather than a fundamental improvement to the Chain-of-Thought mechanism itself. Specifically, it samples multiple CoT traces and applies a series of selection mechanisms, which essentially makes it a variant of the CoT-SC method.
> > > 3.  **Specific Cost Comparison:** Consequently, the authors should provide a comparison of **inference-time token costs** between ARM, ARM+MP, and baselines like CoT-SC to better evaluate the cost-performance trade-off.

---

> > > > ### Author Response · Authors · 2025-11-26
> > > > **Response to new comments by Reviewer dWxr**
> > > >
> > > > [1/3]
> > > > We thank the reviewer for re-reading the paper and acknowledging the validity of the paper's motivation. We are glad that the core motivation is now clear. Below we address the new comments:
> > > >
> > > > ### Response to Request-1:
> > > > We appreciate the reviewer’s question which provides an opportunity for a clarification. Our statement that ARM is _orthogonal_ to MaAS refers to the level of granularity at which ARM and MaAS operate.
> > > > - ARM discovers a granular reasoning operator (a replacement for a single CoT operator step)
> > > > - MaAS performs architectural search over graph of higher level “_agentic operators_”.
> > > >
> > > > Thus, MaAS optimizes a topology over high-level blocks, whereas ARM optimizes a single step-generator itself. Thus, ARM is a strictly more granular object than the MaAS operator: ARM’s evolution occurs within what MaAS would classify as a single node/operator. Our scaffolded search targets per-step competence rather than global orchestration topology (which is relegated to Meta-Agent). Because of this **granularity mismatch**, integrating ARM into MaAS would not be a matter of simply substituting an operator:
> > > > - MaAS assumes each operator is a self-contained multi-call routine which finishes a given task, whereas ARM is constructed by evolving internally coordinated micro-steps that are not directly expressible as a single MaAS-style operator without substantial re-engineering.
> > > > - Moreover, while MaAS supports text-gradient based updates to the operators, these are performed at the level of full operator blocks, each of which encapsulates an entire (LLM) multi-call routine for it’s task. Applying these updates at the granular level that ARM targets is non-trivial: reward is obtained only at the end of complete workflow rollout, and each step-generator would now contain several LLM calls by themselves and yields only the next partial progress. This creates severe **credit assignment** problem, as there is no way to attribute improvements/failures to step-level modifications in the accumulated context over several granular steps, each of which is a multi-call workflow. Our scaffolded objective (Section3.2) is designed **precisely to overcome this problem** by isolating localized windows of reasoning traces to learn a superior refinement of CoT operator.
> > > >
> > > > In summary, ARM enhances the granular CoT step, while MaAS searches over full operator blocks and their topologies. Combining them is non-trivial because MaAS operators and textual gradients are designed for coarse-grained operators, and simple substitution with ARM needs a non-trivial solution to the step-level credit assignment. We leave this for future work.
> > > >
> > > > However, we want to highlight that we have already included comparison with original authors’ MaAS setup based on the reviewer’s earlier comment, which shows that our approach performs better across benchmarks.

---

> > > > ### Author Response · Authors · 2025-11-26
> > > > **[2/3] Response to new comments by Reviewer dWxr**
> > > >
> > > > [2/3]
> > > > ### Response to Request-2:
> > > > We thank the reviewer for closely inspecting the best resulting ARM module (Appendix-E) from the scaffolded search objective (Section-3.2). However, we would like to clarify that recent works on automated agentic system design treats the terms “_agentic workflow_” and “_multi-agentic system_” and “_workflow graphs_” as interchangeable terms.
> > > >
> > > > Following prior work [ADAS, AFlow, MaAS], we define a _multi-agentic system_ (MAS) (or alternatively an _agentic-workflow_) as a graph with a set of LLM invoking nodes (holding a prompt, role etc.) and edges representing communication topology. Within this setup, a CoT operator is a one node graph invoking an LLM only once, a CoT-SC operator has $n$ nodes solving the same task and directing the outputs to a majority voting node.
> > > >
> > > > Both ADAS and AFlow papers follow this definition of agents and justify how code representation of these workflow graphs (MASes) provides advantages such as ability to support explicit control-flow, compositionality, and modularity of LLM-calling nodes with maximal expressivity. Automatic discovery methods such as ADAS, AFlow and MaAS search for more complex workflows constructed from building block operators (See Section-2 of ADAS, and Section-3 of AFlow papers). MaAS also uses code representation for operators but picks the higher order edges between them via a learned controller neural network. The appendices of these papers also show the final agentic workflow artifacts that their search discovers.
> > > >
> > > > Under this standard formulation, the fact that the discovered ARM module is implemented as a “_workflow_” is **not** a distinguishing property: all agentic systems in recent literature are workflow-style DAGs of LLM calls. Chain of Thought reasoning forms the core reasoning engine in all MAS systems, whether manually created or automatically generated. The difference lies in **which workflow** is being optimized, at **what granularity**, and by **what objective**.
> > > >
> > > > Our contribution specifically lies in improving the fundamental CoT level per-step operator. ARM is optimized at the intra-step level via the scaffolded training objective (Section-3.2) which replaces short windows of a full CoT trace, and provides localized credit assignment for each candidate module. This yields a **granular reasoning MAS** (or _agentic workflow_) with behaviors far beyond CoT-SC’s simple fixed n-parallel samples. These include adaptive _chain generation_, _confidence-weighted consensus_, _verification-gated acceptance_, and _selective-refinement_-- all automatically discovered through reflection guided search, and empirically shown to reduce per-step error (Section-6, Appendix-C.4).
> > > >
> > > > Furthermore, ARM preserves the output behavior of CoT (Appendix-D), making it a principled refinement of CoT operator, which can be used as a drop-in replacement at a granular level.
> > > >
> > > > **References**:
> > > >
> > > > ADAS: Automated Design of Agentic Systems - Hu et.al ICLR-2025
> > > >
> > > > AFlow: AFlow: Automating Agentic Workflow Generation - Zhang et.al - ICLR 2025
> > > >
> > > > MaAS: Multi-agent Architecture Search via Agentic Supernet - Zhang et.al - ICML-2025

---

> > > > ### Author Response · Authors · 2025-11-26
> > > > **[3/3] Response to new comments from Reviewer dWxr**
> > > >
> > > > ### Response to Request-3:
> > > >
> > > > We can provide the cost comparison to CoT-SC, in fact it can be computed theoretically. For CoT-SC with $n$ parallel rollouts, the cost-multiple is at least $n$, and thus **scales linearly** while returning diminishing returns [4] (even if discounting the consensus cost with static code for majority voting).
> > > >
> > > > Thus, in our setting with $n=12$, we have a cost multiple of 12, with average performance improvement of +3.9% (See Table-1). Thus, the normalized improvement factor is 0.325. The updated performance gain per cost multiple table is provided below:
> > > >
> > > > | Approach            | Cost multiple (cost) | Performance gain | Gain / Cost multiple |
> > > > |---------------------|----------------------|------------------|-----------------------|
> > > > | CoT                 | 1.0 (0.13 $)         | –                | –                     |
> > > > | CoT-SC (n=12)       | 12                   | +3.9             | 0.325                 |
> > > > | ADAS                | 4.2                  | -2.9             | -0.69                 |
> > > > | AFLow               | 3.5                  | -1.7             | -0.48                 |
> > > > | ARM (Ours)          | 16.6                 | +6.1             | +0.36                 |
> > > > | ARM+Meta (Ours)     | 74.8                 | +11.2            | +0.14                 |
> > > >
> > > >
> > > > We will include these terms for all the other baseline operators as well as CoT-SC with varying $n$ in the final manuscript.
> > > >
> > > > For ARM and ARM+Meta, the improvements are larger (+6.1% and +11.2%), and the normalized gain for ARM-only configuration is higher than for CoT-SC, while adding Meta policy yields substantial performance improvement at a lower normalized gain. Interestingly, the discovered final ARM module (Appendix-E) adopts _adaptive sampling_ and _early stopping_, whereas CoT-SC always pays the $n$ fold cost.
> > > >
> > > > We once again emphasize that while this work focused on performance, our approach can be extended to allow for cost-optimized ARM variants (see lines 739-741)  by accounting for cost in the reviewer agent.
> > > >
> > > > **References:**
> > > >
> > > > [4]: Self-Consistency Improves Chain of Thought Reasoning in Language Models - Wang et.al ICLR-2023

---

> > > > > ### Comment · Reviewer_dWxr · 2025-11-28
> > > > >
> > > > > I thank the authors for answering my questions. While I acknowledge that ARM is significantly more complex than the standard CoT-SC, I maintain the view that it remains fundamentally a variant of CoT/CoT-SC. This is similar to other works (e.g., [1]) that propose generalizable improvements to CoT/CoT-SC. The primary distinction here is that while previous works manually designed these improvements, this work discovers them via search.
> > > > >
> > > > > Looking at the final discovered result, it is essentially CoT augmented with **Adaptive Chain Generation**, **Confidence-weighted Consensus**, **Verification-gated Acceptance**, and **Selective-refinement**. However, these specific ideas have already been proposed in prior literature or share similar philosophies with works such as Adaptive-Consistency [1], Confidence-Informed/Weighted Self-Consistency [2], Self-Verification [3], and Self-Refine [4]. Therefore, I perceive a limited novelty in terms of the discovered mechanisms themselves.
> > > > >
> > > > > Could the authors clarify on which specific benchmarks the current cost comparison results were achieved? Furthermore, to truly demonstrate the advantage of ARM over these existing manual improvements, I believe it is necessary to:
> > > > > 1.  Conduct experiments with a wider range of $N$ (sample sizes) for the baselines.
> > > > > 2.  Compare directly against the specific methods mentioned above ([1], [2], [3]) to show if the discovered combination is indeed superior to manual SOTA designs.
> > > > >
> > > > > In light of the current results and clarifications, I will raise my score to 4.
> > > > >
> > > > > **References:**
> > > > >
> > > > > [1] Aggarwal P, Madaan A, Yang Y. Let’s Sample Step by Step: Adaptive-Consistency for Efficient Reasoning and Coding with LLMs[C]//Proceedings of the 2023 Conference on Empirical Methods in Natural Language Processing. 2023: 12375-12396.
> > > > >
> > > > > [2] Taubenfeld A, Sheffer T, Ofek E, et al. Confidence improves self-consistency in llms[J]. arXiv preprint arXiv:2502.06233, 2025.
> > > > >
> > > > > [3] Weng Y, Zhu M, Xia F, et al. Large language models are better reasoners with self-verification[C]//Findings of the Association for Computational Linguistics: EMNLP 2023. 2023: 2550-2575.
> > > > >
> > > > > [4] Madaan A, Tandon N, Gupta P, et al. Self-refine: Iterative refinement with self-feedback[J]. Advances in Neural Information Processing Systems, 2023, 36: 46534-46594.

---

> > > > > > ### Author Response · Authors · 2025-11-28
> > > > > > **[1/2] Response to latest comments.**
> > > > > >
> > > > > > [1/2] We thank the reviewer for raising the score and their continued engagement. We appreciate the observation that the discovered ARM module integrates mechanisms akin to adaptive consistency, verification, and self-refinement.
> > > > > >
> > > > > > While we agree that the resulting ARM artifact shares components with manual-designs, we respectfully posit that characterizing the work as limited novelty overlooks the **primary contributions of this work**: i) The automated designed itself (Methodology) and ii) The shift from global to atomic granularity (Architecture).
> > > > > >
> > > > > > i) **Designer is a core contribution (Validated by ICLR & ICML Orals)**: The primary contribution of this work is not the final python script it produces, but the Reflection-Guided Evolutionary Search framework that discovers it. The value of such “Automated Design” paradigm, i.e., specifically **searching over a space of known operator strategies** (CoT-SC, Refine, Critique etc.) to **stitch together a final workflow** is well-established by the field’s most recent seminal works. These Oral papers at ICLR and ICML, along with ADAS (ICLR) [3], demonstrate that the community is interested in such automated workflow generation techniques:
> > > > > >
> > > > > > - **AFlow (ICLR-2025 Oral)**: As noted in related work, automated frameworks are developed because constructing these systems requires significant manual effort. Configuring individual operators (voting, verification, refinement etc.) into an optimal workflow is non-trivial; it faces a combinatorial explosion of possible topologies, along with challenges in ensuring format-correctness, goal fidelity etc. As AFlow authors note, manually hand-crafting such workflows (MASes) “_requires significant human effort, which limits the scalability and adaptability_” [1] (quoted verbatim). Thus, manual construction requires extensive trial-and-error to find a solution that does not effectively downgrade performance at the domain level. (Note that ARM goes beyond this paradigm and shows cross-domain generalization as shown in Table-1)
> > > > > >
> > > > > > - **MaAS (ICML 2025 Oral)**: Similarly, MaAS authors note the challenge that constructing these systems often requires "_labor-intensive manual designs_" [2] (quoted verbatim) and domain-specific tuning.
> > > > > >
> > > > > > Our work operates on this same core premise: finding the optimal configuration of these known operator strategies is a complex search problem (over prompts, topologies, hyperparameters such as confidence threshold etc.) and risks human biases. Empirically, our results (Table 1) confirm that the automated designer finds a superior configuration: ARM consistently outperforms the manual baselines (CoT-SC, Self-Refine, Debate) that utilize some of the same underlying primitives.
> > > > > >
> > > > > > The fact that our search converges on robust patterns (like consensus, verification etc.), in fact validates the search algorithm's effectiveness – showing that it can independently navigate the vast code space and combine known agentic strategies into one coherent workflow for a granular reasoning operation. AFlow paper in particular notes that automatic discovery paradigm has the “_ability to reduce human effort by evolving from an empty workflow to a solution **highly similar** to manually designed workflows_” [1] (quoted verbatim from Appendix-B.1 of that paper).
> > > > > >
> > > > > > ii) **Granularity-Recursive (atomic) vs Global**: Regarding the artifact itself, ARM is structurally distinct from prior manual methods (CoT-SC, Self-Refine etc.) and automatic methods (ADAS, AFlow, MaAS).
> > > > > >
> > > > > > - Prior methods operate at the global/problem level (e.g. generate full answer -> vote/refine etc.)
> > > > > > - ARM encapsulates these agentic behaviors into a single atomic reasoning step, applied recursively.
> > > > > >
> > > > > > By pushing the “_agentic workflow_” paradigm down to the step-generator module level, ARM corrects reasoning errors before they cascade (Section-6, Appendix-C.4). This recursive application of MAS (or agentic-workflow) represents a new **granular architectural paradigm** (Agentic-operation-workflow rather than just Agentic-Workflow).
> > > > > >
> > > > > > ---
> > > > > >
> > > > > > **Reg clarification on the cost-comparison**: We used the average performance and average inference cost across the four benchmarks considered in Section-4.1 (AIME, HMMT, GPQA, Livebench) for the best performing model (GPT-4.1-nano). This represents a balanced, overall view of the cost-performance analysis and shows that ARM module has higher performance compared to all baselines, as well as best performance gain per cost multiple. When ARM is combined with the meta-policy, it achieves the highest performance at a higher cost.

---

> > > > > > ### Author Response · Authors · 2025-11-28
> > > > > > **[2/2] Response to latest comments**
> > > > > >
> > > > > > **Reg wider range of N for self-consistency:** We have already picked quite a large N(=12) for the baseline, which is much stronger than the baselines used in recent literature (Both MaAS and AFlow use n=5). Nonetheless, we are happy to include an ablation on the baseline’s hyperparameter choice of N in the final draft. However, we want to emphasize that it is well noted in literature that self-consistency approach has diminishing returns with increasing N and the cost scales linearly [4]
> > > > > >
> > > > > > ---
> > > > > >
> > > > > > **Regarding comparison with [5,6,7]:** We respectfully posit that the three cited works represent specific manual optimizations of two fundamental reasoning primitives: Consensus and Refinement. As noted above, we have already compared with the strongest versions of these core reasoning baselines. This is consistent with prior literature [1,3], which also shows the Debate, CoT-SC and Self-Refine to have the top performance among manual methods (ranking among these three varies by dataset). We believe that requiring comparisons against every minor manual variant shifts the focus away from the core contribution: an automated method that synthesizes agentic workflows at the granular reasoning step level.
> > > > > >
> > > > > > Nonetheless, we have started running these experiments and will include these baselines in the final draft.
> > > > > >
> > > > > >
> > > > > > ---
> > > > > >
> > > > > > *References:*
> > > > > >
> > > > > > [1]: AFlow: AFlow: Automating Agentic Workflow Generation - Zhang et.al - ICLR 2025
> > > > > >
> > > > > > [2]: MaAS: Multi-agent Architecture Search via Agentic Supernet - Zhang et.al - ICML-2025
> > > > > >
> > > > > > [3]: ADAS: Automated Design of Agentic Systems - Hu et.al ICLR-2025
> > > > > >
> > > > > > [4]: Self-Consistency Improves Chain of Thought Reasoning in Language Models - Wang et.al ICLR-2023
> > > > > >
> > > > > > [5]: Aggarwal P, Madaan A, Yang Y. Let’s Sample Step by Step: Adaptive-Consistency for Efficient Reasoning and Coding with LLMs[C]//Proceedings of the 2023 Conference on Empirical Methods in Natural Language Processing. 2023: 12375-12396.
> > > > > >
> > > > > > [6]: Taubenfeld A, Sheffer T, Ofek E, et al. Confidence improves self-consistency in llms[J]. arXiv preprint arXiv:2502.06233, 2025.
> > > > > >
> > > > > > [7]: Weng Y, Zhu M, Xia F, et al. Large language models are better reasoners with self-verification[C]//Findings of the Association for Computational Linguistics: EMNLP 2023. 2023: 2550-2575.

---

### Author Response · Authors · 2025-11-26
**General Response - Summary of Revisions in the Updated Manuscript**

We want to thank the reviewers for their constructive feedback. Most comments centered on clarifying computational cost, convergence analysis, interpretability, and the discovered artifacts. Accordingly, we have expanded the appendix with all the requested analyses and the code modules.

- **Appendix-A:** Added convergence analysis of the ARM module during search (A.1) (requested by reviewer 5da4), and cost analysis of both search and inference (A.2) (requested by all reviewers).
- **Appendix-C:** Strengthened the theoretical section with more rigor to address reviewer T74Q’s concerns (Q3) about the meta-policy learned on CoT and deployed off-policy with ARM.
- **Appendix-D:** Added examples of rollouts from ARM compared to CoT, addressing reviewer 5da4’s concerns on interpretability.
- **Appendix-E:** Added the best ARM module discovered by our search algorithm, as requested by all reviewers.
- **Appendix-F:** Added the best Meta Policy module discovered by our search algorithm, as requested by all reviewers.
- **Appendix-G:** Detailed implementation details for reproducibility. We have also committed to release the full code upon publication.
- **Appendix-H:** An example of reviewer-based mutation of the ARM module during search, directly addressing reviewer 5da4’s concerns on interpretability.

All remaining concerns have been addressed directly in our rebuttal comments. Once again, we want to thank the reviewers for their time. We believe these changes fully resolve all the raised concerns, and we welcome any additional comments.

---

### Comment · Area_Chair_TGHK · 2025-11-27
**Please review the authors' responses and provide feedback ASAP**

Dear Reviewers,

Thank you for your essential contributions to the review process. The authors have submitted their responses to your initial reviews.

I kindly ask you to carefully review the authors' responses for this submission. Your timely assessment of how the authors have addressed your original concerns is a critical step in reaching a final decision.

Please provide your feedback and any necessary updates to your reviews as soon as possible to ensure we can meet our tight schedule for the discussion phase.

Your prompt attention to this matter is highly appreciated.

Regards,

-AC

---

### Author Response · Authors · 2025-11-30
**Summary of discussion phase.**

We acknowledge the new policy reg score and discussion freeze. We thank both ACs for their time during this transition. Below, we summarize how we addressed concerns from Reviewers dWxr, T74Q, and 5da4, detailing our new evidence, revised manuscript, and discussion outcomes.

## Concern-1 [Reviewer- dWxr]: ADAS/AFlow underperform CoT, so baselines are questionable.
Our rebuttal:
- Explained why ADAS/AFlow can underperform CoT on long-horizon reasoning (validation overfitting leads to brittle workflows)
- Cited independent reports from MAS-Zero (NeurIPS W 2025 Oral) and MaAS (ICML 2025 Oral) showing the same ADAS < CoT pattern.
- Clarified that we used the original codebases from the baseline authors with fair evaluations.

**Outcome:** Reviewer dWxr agreed with this, and explicitly acknowledged that they now agree with the validity of the paper’s motivation. They raised their score from 2 to 4 along with additional questions just before discussion freeze.

## Concern-2 [All reviewers] Missing cost analysis – Search and Inference time.
Our rebuttal:
-  Added Appendix A.2 with full numbers including both search and inference time costs.
- ARM represents a "System 2" approach where **each reasoning step is executed by an agentic system**, thus higher cost is expected.
- Amortization analysis showing ARM search is >3x cheaper than ADAS/AFlow when used across 4 domains × 3 models.
- Inference-time cost vs performance table comparing CoT, CoT-SC, ADAS, AFlow, ARM, ARM+Meta
- ARM achieves best gain-per-cost-multiple among recursive methods (+6.1% at 16.6× = 0.36 vs CoT-SC = 0.325). ARM+Meta yields highest raw performance (+11.2).
- Our method automatically extends to exploring cost-optimized variants if cost term is included in the reviewer agent.

**Outcome:** dWxr accepted and requested CoT-SC (highest performing baseline) numbers, which we added.

## Concern-3 [Reviewer- dWxr] Missing MaAS baseline
Our rebuttal:
-  Added MaAS results using GPT-4.1-nano (the best performing model)
- ARM outperformed MaAS on all benchmarks.
- Explained why MaAS is orthogonal (MaAS operates at workflow-level; ARM at per-step operator level).

**Outcome:** Reviewer was satisfied, did not raise additional questions on this point.

## Concern-4 — [Reviewer dWxr] ARM resembles known strategies combined into a workflow.

Our Rebuttal:
- Clarified that all recent automated agentic-system works (ADAS, AFlow, MaAS) define multi-agent systems via workflow graphs; “agentic workflow” is the standard abstraction.
- Automated workflow generation is the core idea of this research line; reusing known strategies is expected (and validated by multiple Orals including ADAS (ICLR 2025), AFlow (ICLR 2025), and MaAS (ICML 2025)). Manual design suffers from a combinatorial explosion of possible workflows.
- Designer is a core contribution (validated by ICLR & ICML Orals). Automated methods intentionally build workflows from known inference strategies—that is the point of this research line, not a weakness. We cited verbatim quotes from prior research as evidence.
- Granularity is the second core contribution. By pushing the “agentic workflow” paradigm down to the step-generator level, ARM corrects errors before they cascade (Section-6, Appendix-C.4), establishing a new granular architectural paradigm.

Outcome: N/A (Discussion freeze).

## Concern-6 — [Reviewer 5da4] “Interpretability + lack of examples for how ARM evolves.”
Our rebuttal:
- Added Appendix D: step-by-step rollouts comparing CoT vs ARM which shows that ARM steps are as interpretable as CoT.
- Added Appendix H: full example of how reviewer-based mutation created better modules.
- Added explicit code artifacts for ARM (Appendix E) and Meta-policy (Appendix F).
- Added Appendix G with other implementation details.

## Reviewers asked for the following information which we provided (see general comment).
- The code for discovered modules (Appendix-E,F)
- Details on Reviewer module and it’s feedback: Provided in Appendix-H.
- Why is l=3 chosen: We provided the theoretical justification (smallest window satisfying 3 kinds of information flows).
- Convergence analysis of optimization process.
- Validity of scaffolded search with CoT surrogate: Expanded theory section (Appendix-C) connecting it to the ablations (Section-6)

## Final Comments:
Based on the above, we believe we have addressed all the concerns of the reviewers. We note that the only reviewer who participated in discussion was also the harshest in their initial review, but acknowledged the validity of the work, and increased score after our clarifications. We believe we have also addressed their newer set of comments satisfactorily, and were anticipating another score raise just before the freeze. The main concerns from other reviewers was also not about the method, or the experiments but more about requiring more information. This was addressed exhaustively with new tables, figures and details in the appendix.

---

### Note · Program_Chairs · 2026-01-17
**Submission Desk Rejected by Program Chairs**

The following references in this submission do not refer to real documents and/or have major errors in bibliographic information:

 Lei Zhang, Feng Xu, Zongyi Yu, Chen Zhu, and Yu Qian. Agent-flow: A flexible and scalable multi-agent platform for real-life tasks. arXiv preprint arXiv:2402.17779, 2024.
Wei Zhang, Chen Liu, Ananya Patel, Ming Zhao, and Jie Huang. Flowreasoner: Automatic multiagent system generation for complex reasoning. arXiv preprint arXiv:2502.08123, 2025c. Accessed: YYYY-MM-DD.
Jean Kaddour, Stephen O'Keeffe, Oscar O'Hara, Joe Arts, Sam Sutherland, William Gillingham, Charlie Lee, Leanne d’Avila de Moraes, Adam de Wynter, and Ali Bibi. LiveBench: A real-time llm benchmark. arXiv preprint arXiv:2311.08277, 2023.
Akshat Dubey. A simple baseline for multi-agent reasoning. arXiv preprint arXiv:2310.02023, 2023.
Sungwoo Kim, Lin Xu, Yifan Guo, Arif Rahman, and Shiyi Wang. Aflow: Automating agentic workflow generation for large language models. In Proceedings of the 2024 Conference on Empirical Methods in Natural Language Processing (EMNLP), 2024. Accessed: YYYY-MM-DD.